# The CD73 immune checkpoint promotes tumor cell metabolic fitness

David Allard[1,2,3], Isabelle Cousineau[1,3], Eric H Ma[4], Bertrand Allard[1,3], Yacine Bareche[1,2,3], Hubert Fleury[1,3], John Stagg[1,2,3]*

[1]Centre de Recherche du Centre Hospitalier l'Université de Montréal, Montreal, Canada; [2]Faculté de Pharmacie, Université de Montréal, Montreal, Canada; [3]Institut du Cancer de Montréal, Montreal, Canada; [4]McGill Goodman Cancer Research Centre, Montréal, Canada

**Abstract** CD73 is an ectonucleotidase overexpressed on tumor cells that suppresses anti-tumor immunity. Accordingly, several CD73 inhibitors are currently being evaluated in the clinic, including in large randomized clinical trials. Yet, the tumor cell-intrinsic impact of CD73 remain largely uncharacterized. Using metabolomics, we discovered that CD73 significantly enhances tumor cell mitochondrial respiration and aspartate biosynthesis. Importantly, rescuing aspartate biosynthesis was sufficient to restore proliferation of CD73-deficient tumors in immune deficient mice. Seahorse analysis of a large panel of mouse and human tumor cells demonstrated that CD73 enhanced oxidative phosphorylation (OXPHOS) and glycolytic reserve. Targeting CD73 decreased tumor cell metabolic fitness, increased genomic instability and suppressed poly ADP ribose polymerase (PARP) activity. Our study thus uncovered an important immune-independent function for CD73 in promoting tumor cell metabolism, and provides the rationale for previously unforeseen combination therapies incorporating CD73 inhibition.

## Editor's evaluation

This important study demonstrates a non-canonical, cancer-cell intrinsic role of the ectonucleotidase CD73 in the regulation of cancer cell metabolism. The evidence supporting the claims is convincing and of high quality.

*For correspondence:
john.stagg@umontreal.ca

## Introduction

CD73 is an ectonucleotidase that catalyzes the phosphohydrolysis of adenosine monophosphate (AMP), thus contributing to the breakdown of pro-inflammatory adenosine triphosphate (ATP) into immunosuppressive ADO. Accumulation of extracellular ADO in the tumor microenvironment favors cancer progression by activating ADO specific G-coupled protein receptors (GPCRs) expressed on immune cells, endothelial cells, fibroblasts and tumor cells. Of the four ADO receptors, high affinity A2A and low affinity A2B receptors elevate intracellular cyclic AMP (cAMP) levels and contribute to suppress immune cell function. In human cancers, such as triple negative breast cancer (TNBC) and ovarian cancer, CD73 expression is generally associated with poor clinical outcomes, impaired tumor immune surveillance and resistance to chemotherapy (*Loi et al., 2013*; *Bareche et al., 2021*). Potent and selective CD73 inhibitors have recently been developed and are now being evaluated in randomized clinical trials with encouraging preliminary results reported in lung cancer and pancreatic cancer patients (*Allard et al., 2020*; *Augustin et al., 2022*).

While CD73 inhibitors can stimulate anti-tumor immunity, early studies reported that targeting CD73 can also inhibit human tumor xenografts in severely immunocompromised mice (*Zhou et al.,*

*2007*; *Zhi et al., 2007*; *Zhi et al., 2010*; *Rust et al., 2013*). Moreover, pharmacological inhibition or gene-deletion of CD73 has also been observed to increase the sensitivity of CD73-expressing tumor cells to cytotoxic drugs in vitro (*Loi et al., 2013*; *Ujházy et al., 1996*; *Yu et al., 2021*). Although activation of pro-survival pathways downstream of A2B receptors has been proposed to explain immune-independent effects of CD73, alternative hypotheses have not been explored (*Mittal et al., 2016*; *Lan et al., 2018*).

Rewiring of cancer cells' metabolism is central to their adaptation to hostile tumor microenvironments (*Zhi et al., 2010*). While cancer cells can shift their metabolism from highly efficient ATP-producing oxidative phosphorylation (OXPHOS) to low ATP-yielding glycolysis despite presence of oxygen (e.g. Warburg effect), mitochondrial respiration has been shown to be critical for proliferating cells and for maintaining tumor cells invasiveness, metabolic adaptation, chemoresistance and stemness. Accordingly, tumor cells are not only capable of respiration but often require respiration for survival (*Ashton et al., 2018*).

In proliferating cells, mitochondrial respiration provides electron acceptors to support biosynthesis of aspartate, an indispensable amino acid for nucleotide and protein synthesis. Mitochondrial respiration requires nicotinamide adenine dinucleotide (NAD) in its oxidized and reduced forms to allow movement of electrons through the electron transport chain (ETC). In addition of supporting respiration, NAD is also an important co-substrate for many enzymes, notably poly-ADP-ribose polymerases (PARPs) that play a central role in genomic stability by catalyzing the addition of poly-ADP-ribose to proteins (also known as PARylation; *Wei and Yu, 2016*; *Pommier et al., 2016*).

Intracellular NAD levels are maintained either by de novo synthesis from L-tryptophan (Trp), or by more effective salvage pathways that recycle NAD from nicotinamide (NAM), nicotinamide mononucleotide (NMN), or nicotinamide riboside (NR) (*Verdin, 2015*). Extracellular NMN is dephosphorylated to NR before cellular internalization by equilibrative nucleotide transporters (ENTs). Intracellular NR is then re-phosphorylated by NRK1 into NMN, a substrate for adenylyl transferases (NMNATs) to generate NAD (*Ratajczak et al., 2016*; *Katsyuba et al., 2020*). NMN can also be generated from intracellular or extracellular NAM via the rate-limiting enzyme NAMPT (*Katsyuba et al., 2020*). Interestingly, some studies have suggested a role for CD73 in NAD synthesis from extracellular NR (*Garavaglia et al., 2012*; *Mateuszuk et al., 2020*). Yet, the impact of CD73 on tumor cell metabolic pathways remains unknown (*Grozio et al., 2013*).

We here investigated metabolic consequences of targeting CD73 in a panel of human and murine cancer cells. We observed that CD73 deficiency or pharmacological inhibition induces an important aspartate-dependent metabolic vulnerability in tumor cells characterized by suppressed mitochondrial respiration and increased genomic instability. Our study thus provides a rationale for exploring novel combination strategies with anti-CD73 therapies, such as PARP inhibition and mitochondria targeting agents.

## Results

### CD73 favors tumor growth independently of its immunosuppressive function

We evaluated the impact of tumor cell-associated CD73 by transiently transfecting a CRISPR-Cas9 construct into human and mouse tumor cell lines followed by flow cytometry-based sorting of polyclonal CD73-negative (CD73neg) and CD73-positive (CD73pos) fractions (*Figure 1—figure supplement 1A–F*). Consistent with prior work (*Rust et al., 2013*) and in support of immune-independent effects, CD73 deletion in MDA-MB-231 human triple negative breast cancer (TNBC) cells significantly delayed tumor growth in severely immunodeficient Nod-Rag-gamma (NRG) mice, which lack T cells, B cells, NK cells and functional macrophages (*Figure 1A*). To further characterize the immune-independent effects of CD73, we next compared the in vitro proliferation rates of CD73pos and CD73neg MDA-MB-231 cells. While *NT5E* (gene encoding for CD73) gene deletion had no impact in standard cell culture conditions (i.e. 25 mM glucose and 2 mM glutamine), it significantly suppressed cell proliferation and viability in nutrients-limiting conditions (*Figure 1B*, *Figure 1—figure supplement 2*), despite no difference in cellular uptake of glucose or glutamine, as shown using a fluorescent glucose analog (i.e. 2-NDBG; *Figure 1C*) and by measurements of glutamine depletion in media (*Figure 1D*).

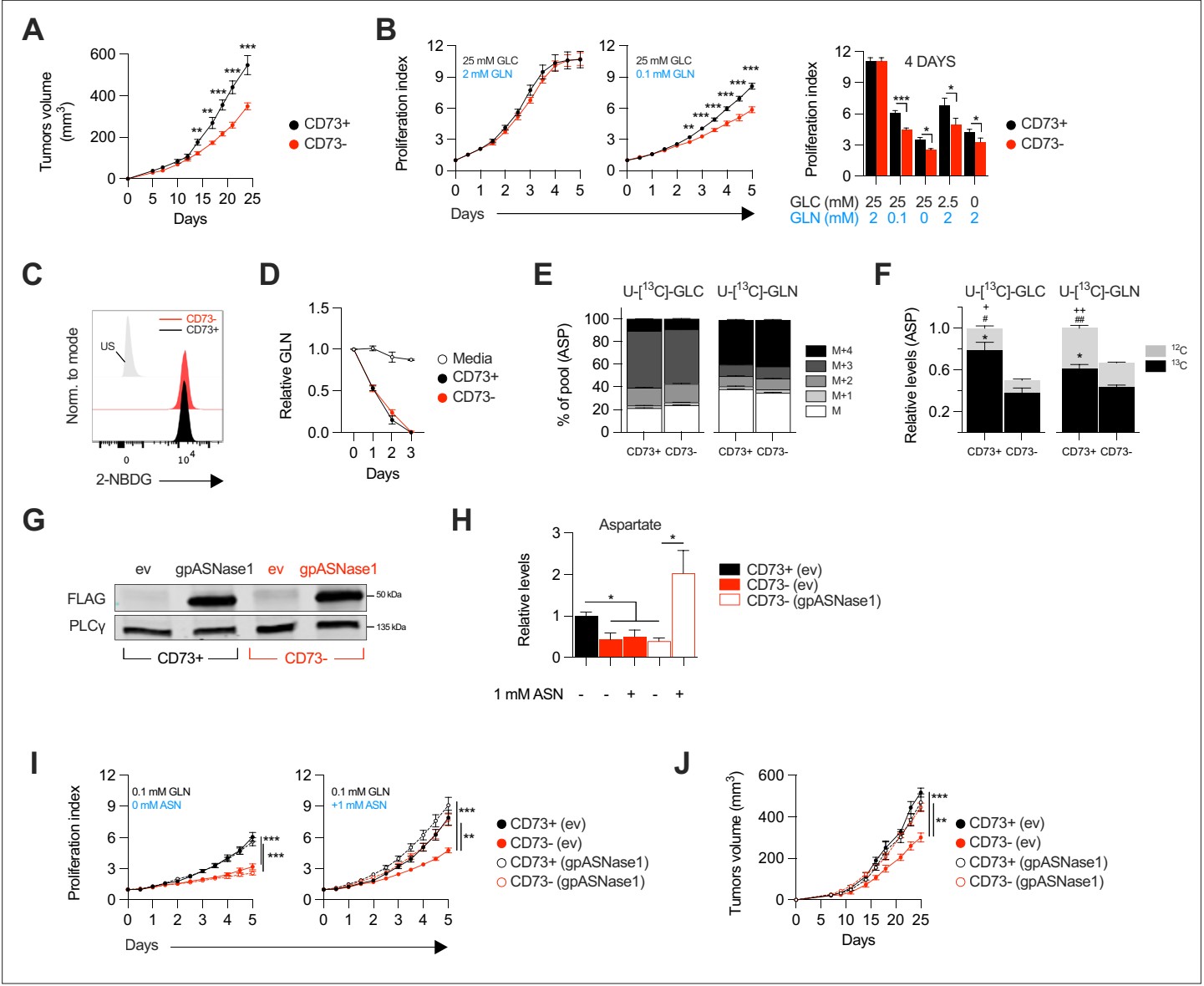

**Figure 1.** CD73-deficiency delays tumor growth independently from immune suppression through reduced aspartate biosynthesis. (**A**) Sub-cutaneous tumor growth of CD73pos (CD73+) and CD73neg (CD73-) MDA-MB-231 tumors in immune-deficient NRG mice. Results shows combined data of two independent experiments cumulating 15 mice per groups. (**B**) In vitro proliferation of CD73 + and CD73- MDA-MB-231 cells was analyzed by Incucyte live imaging technology in presence of excess or restricted conditions of glucose (GLC) and glutamine (GLN). Data from representative proliferation curves (left panels) and pooled quantification of proliferation after 4 days of culture (right panel) are shown (n=3). (**C**) FACS analysis of in vitro incorporation of a fluorescent glucose analog (2-NBDG) in CD73 + and CD73- MDA-MB-231 cells (n=3). US = unstained control. (**D**) Glutamine depletion assay in culture media of MDA-MB-231 CD73 + and CD73- cells measured using a commercial kit (Promega). Cell-free media was used as a control. (**E**) Mass isotopomer distribution (MID) of U-[$^{13}$C]-glucose-derived (U-[$^{13}$C]-GLC; left panel) or U-[$^{13}$C]-glutamine-derived (U-[$^{13}$C]-GLN; right panel) aspartate compared between CD73 + and CD73- MDA-MB-231 cells (n=1). M represents the pool of unlabeled aspartate. M+1, M+2, M+3 and M+4 represent the pools of isopotologues labeled with $^{13}$C carbon. (**F**) SITA analysis of glucose (left panel) and glutamine (right panel) contribution to ASP intracellular biosynthesis (n=1). ASP levels were normalized on cell number and are shown relative to CD73 + cells. Stats: *comparison of $^{13}$C levels; #Comparison of $^{12}$C levels; + Comparison of total $^{13}$C+$^{12}$C levels. (**G**) Western blot of CD73 + and CD73- MDA-MB-231 cells expressing empty vector (ev) or FLAG-tagged gpASNase1. PLCγ is shown as a loading control. (**H**) Intracellular ASP levels measured in cells expressing or not (ev) the gpASNase1 upon exposure to exogenous ASN (1 mM) for 24 h. ASP levels were normalized on cell number and are shown relative to CD73+ (ev) cells (n=1). (**I**) In vitro proliferation of gpASNase1-expressing and non-expressing (ev) MDA-MB-231 cells in presence (right panel) or not (left panel) of exogenous ASN (n=2). (**J**) In vivo tumor growth of 2×10$^6$ gpASNase1-expressing or not (ev) CD73 + and CD73- MDA-MB-231 cells in NRG mice (CD73+ (ev) n=5; CD73-(ev) n=7; CD73+ (gpASNase1) n=8; CD73- (gpASNase1) n=8). Means +/- SEM are shown (*p<0.05; **p<0.01; ***p<0.001 by Student T tests).

The online version of this article includes the following source data and figure supplement(s) for figure 1:

*Figure 1 continued on next page*

*Figure 1 continued*

**Source data 1.** Original raw unedited blots and uncropped blots with the relevant bands labeled that composes *Figure 1G*.

**Source data 2.** Stable isotope tracing analysis (SITA) raw data for both U-[$^{13}$C]-glucose and U-[$^{13}$C]-glutamine tracing in MDA-MB-231 CD73 +and CD73- cells that composes *Figure 1E–F* and *Figure 1—figure supplement 3*.

**Figure supplement 1.** Generation of CD73-knockout MDA-MB-231 cell line.

**Figure supplement 1—source data 1.** Original raw unedited blots and uncropped blots with the relevant bands labeled that composes *Figure 1—figure supplement 1B, D*.

**Figure supplement 2.** In vitro proliferation and viability analysis of CD73 + and CD73- MDA-MB-231 cells.

**Figure supplement 3.** Glucose and glutamine SITA and transcriptional analysis of TCA cycle enzymes in MDA-231 cells.

**Figure supplement 4.** CD73 expression and enzymatic activity in gpASNase1-expressing MDA-MB-231 cells.

## CD73-deficiency impairs aspartate biosynthesis and tumor growth

We hypothesized that CD73 was involved in promoting tumor cell metabolism. To test this, we performed stable isotope tracer analysis (SITA) of labeled glucose or glutamine in CD73pos and CD73neg MDA-MB-231 cells. Strikingly, while the isopotomers distribution and relative abundance of most intermediate metabolites were unaltered (*Figure 1—figure supplement 3*), aspartate biosynthesis from glucose or glutamine (*Figure 1E*) was significantly reduced – by nearly 50% – in CD73neg tumor cells (*Figure 1F*).

To assess whether this decrease in aspartate biosynthesis was responsible for the suppressed growth of CD73-deficient tumor cells, we tested the ability of aspartate supplementation to restore proliferation of CD73neg MDA-MB-231 cells. We used a strategy developed by *Sullivan et al., 2018*, whereby an asparaginase enzyme (i.e. gpASNase1) is introduced into cells to restore intracellular aspartate levels upon exposure to exogenous asparagine (since exogenous aspartate cannot be readily incorporated into cells). Stable expression of gpASNase1 (*Figure 1G*) did not affect CD73 expression or enzymatic activity (*Figure 1—figure supplement 4*). Asparagine treatment of gpASNase1-expressing cells indeed significantly increased intracellular aspartate levels (*Figure 1H*) and effectively restored proliferation of CD73neg MDA-MB-231 tumor cells, both in vitro in glutamine-restricted conditions (*Figure 1I*) and in vivo in immunodeficient NRG mice (*Figure 1J*). Taken together, our results demonstrated that CD73-deficiency impaired tumor growth via impaired aspartate biosynthesis.

## CD73 regulates metabolic fitness of cancer cells

Since aspartate biosynthesis is an essential function of mitochondrial respiration (*Birsoy et al., 2015*; *Garcia-Bermudez et al., 2018*; *Sullivan et al., 2015*), and that aspartate fuels OXPHOS (*Cheng et al., 2018*), we evaluated the impact of CD73 on mitochondrial respiration. For this, we measured oxygen consumption rate (a surrogate of OXPHOS) and extracellular acidification rate (a surrogate of glycolysis) in response to standard mitochondrial stress tests (*Figure 2A*). Strikingly, CD73neg MDA-MB-231 cells displayed significantly reduced OXPHOS (*Figure 2B*) and reduced glycolytic reserve compared to CD73pos cells in both glutamine replete and glutamine limiting conditions (*Figure 2C*, *Figure 2—figure supplement 1*). In addition, blocking CD73 with a neutralizing mAb (*Häusler et al., 2014*) also significantly suppressed OXPHOS and glycolytic reserve (*Figure 2B–C*). Importantly, similar results were obtained in additional murine cell lines (4T1, SM1 LWT1), human cell lines (SK23MEL, PANC1, SKOV3, and HCC1954) and in primary murine embryonic fibroblasts (MEF) derived from CD73-deficient mice (*Figure 2D–H*, *Figure 2—figure supplement 2A–B*).

Consistent with the observed defect in mitochondrial respiration, CD73 *NT5E* gene-targeted tumor cells also displayed reduced ratios of ATP to ADP or AMP (*Figure 2I*, *Figure 2—figure supplement 3A*). However, no difference was observed in *GOT1* and *GOT2* mRNA expression, which encode enzymes that generate aspartate from oxaloacetate (*Birsoy et al., 2015*; *Figure 2—figure supplement 3B*), or other tricarboxylic acid (TCA) cycle-related enzymes (*Figure 2—figure supplement 3B*).

As CD73 is overexpressed in tumor-initiating cells (TIC) to promote stemness (*Katsuta et al., 2016*; *Song et al., 2017*), and that TIC rely on OXPHOS to support tumorigenesis (*Snyder et al., 2018*), we next evaluated the impact of CD73 on the metabolic fitness of putative TIC (ALDH$^{High}$ CD44$^+$ CD24$^-$) derived from MDA-MB-231 cell cultures (*Figure 2—figure supplement 2C*). Similar to the bulk culture, CD73neg TIC also displayed significantly suppressed OXPHOS and reduced glycolytic reserve

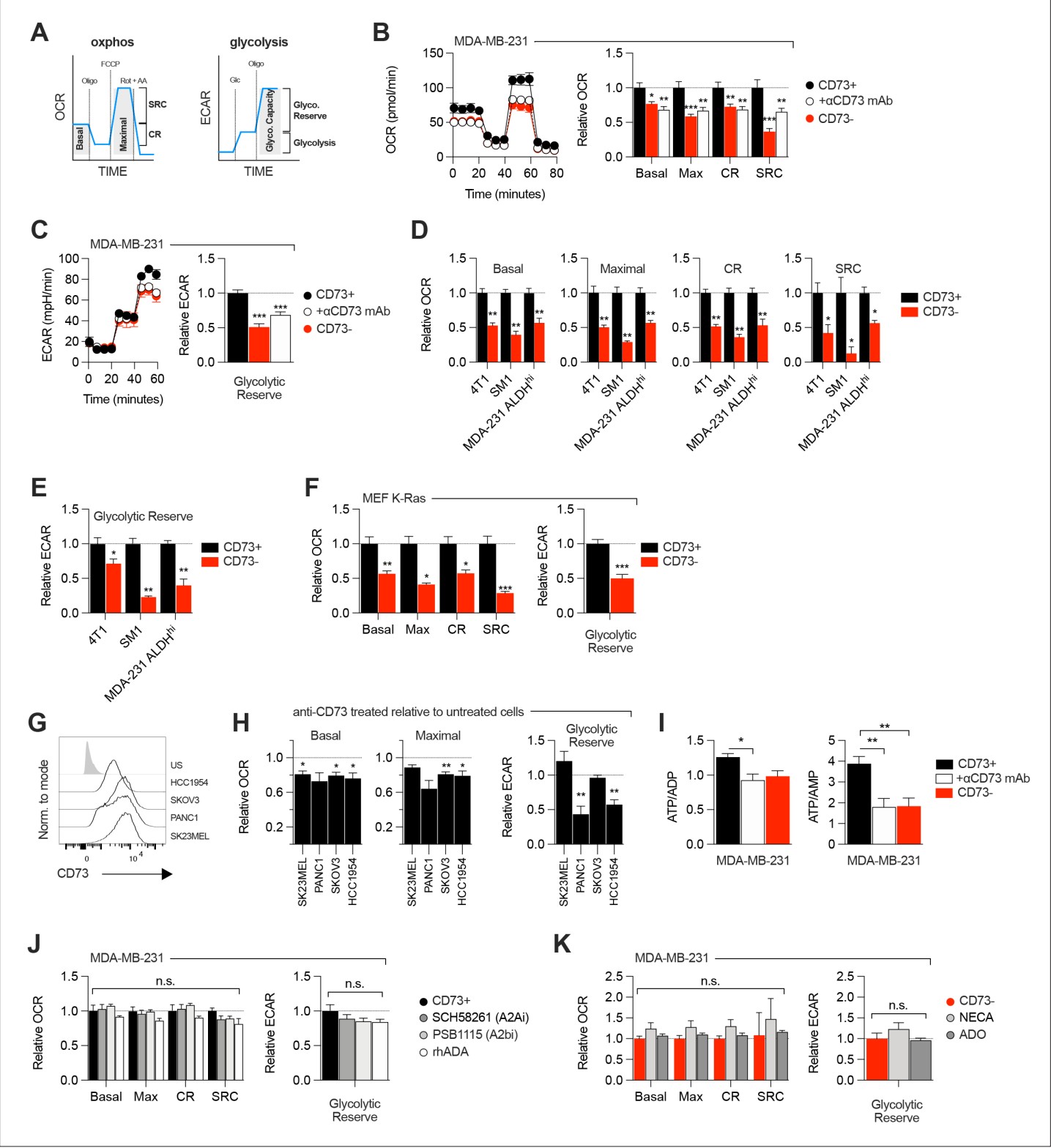

**Figure 2.** CD73 regulates metabolic fitness of cancer cells independently from adenosine signaling. (**A**) Schematized metabolic profiling and parameters generated on a Seahorse analyzer. OXPHOS and glycolytic parameters were calculated using Agilent calculator tool. Rot = rotenone, AA = antimycin A, Oligo = oligomycin, Glc = glucose, OCR = oxygen consumption rate, ECAR = extracellular acidification rate, SCR = spare respiratory capacity which represents the difference between maximal OCR and basal OCR (indicator of cell fitness/flexibility), CR = coupled-respiration which represents the decrease of basal respiration upon inhibition of ATP synthase (indicator of basal respiration used for ATP production to meet energetic needs of the

*Figure 2 continued on next page*

*Figure 2 continued*

cells). (**B**) OXPHOS profile (left panel) and parameters (right panel) of CD73pos (CD73+;+/-1 µg/mL anti-CD73 mAb 7G2 for 48 hr) and CD73neg (CD73-) MDA-MB-231 cells. OXPHOS parameters are shown relative to CD73 + cells from pooled experiments (n=3). (**C**) Glycolytic profile (left panel) and glycolytic reserve (right panel) of CD73+ (+/-1 µg/mL anti-CD73 mAb 7G2 for 48 hr) and CD73- MDA-MB-231 cells. Glycolytic reserve is shown relative to CD73 + cells from pooled experiments (n=3). (**D–E**) Oxphos parameters (**D**) and glycolytic reserve (**E**) compared between various CD73-expressing and CD73 CRISPR-knockout cell lines. ALDHhi cells were sorted from MDA-MB-231 cells using the Aldefluor kit. Data are shown relative to CD73 + cells (MDA-231 ALDHhi n=1, 4T1 and SM1 LWT1 n=2). (**F**) OXPHOS parameters (left panel) and glycolytic reserve (right panel) compared between primary MEF isolated from WT and CD73$^{-/-}$ C57BL/6 mice and transformed with an oncogenic K-Ras. Data are shown relative to WT cells (n=2). (**G**) CD73 expression profile on various human cancer cell lines analyzed by FACS. US = unstained control. (**H**) OXPHOS parameters (left and middle panels) and glycolytic reserve (right panel) of anti-CD73 mAb (7G2)-treated cells compared to untreated cells. Data are shown relative to respective untreated cells (n=1 per cell line). (**I**) LC-MS analysis of total intracellular levels of ATP/ADP and ATP/AMP ratios in CD73+ (+/-1 µg/mL anti-CD73 mAb 7G2 for 48 h) and CD73- MDA-MB-231 cells (n=2). (**J**) OXPHOS parameters (left panel) and glycolytic reserve (right panel) of CD73 + MDA MB-231 cells treated with either A2A inhibitor SCH 58261, A2B inhibitor PSB 1115 or human recombinant ADA. Data are shown relative to untreated CD73 + cells (A2ai and A2bi n=2, rhADA n=1). (**K**) OXPHOS parameters (left panel) and glycolytic reserve (right panel) of NECA- or ADO-treated CD73- MDA-MB-231 cells. Data are shown relative to untreated CD73- cells (NECA n=2, ADO n=1). Means +/- SEM are shown (*p<0.05; **p<0.01; ***p<0.001 by Student T tests).

The online version of this article includes the following source data and figure supplement(s) for figure 2:

**Figure supplement 1.** CD73-deficient MDA-MB-231 cells retain impaired metabolic flexibility in glutamine-limiting conditions.

**Figure supplement 2.** Generation of CD73-knockout cancer cell lines.

**Figure supplement 3.** Intracellular ATP, ADP and AMP metabolites measurement and transcriptional analysis of metabolic enzymes in MDA-MB-231 CD73 +and CD73- cells.

**Figure supplement 3—source data 1.** Raw microarray data that composes *Figure 2—figure supplement 3B*.

compared to CD73pos TIC (*Figure 2D–E*). Taken together, our data demonstrated that CD73 plays a critical role in promoting mitochondrial respiration in proliferating tumor cells.

## ADO-independent metabolic functions of CD73

We next evaluated whether the metabolic functions of CD73 were mediated by ADO receptor signaling. MDA-MB-231 cells express A2A and A2B receptors (*Figure 1—figure supplement 1F*; *Fernandez-Gallardo et al., 2016*). Nevertheless, treatment with an A2A antagonist (SCH58261) or an A2B antagonist (PSB1115) had no effect on mitochondrial respiration or glycolysis (*Figure 2J*). To rule-out a potential role for other ADO receptors or ADO intracellular uptake, we treated MDA-MB-231 cells with recombinant adenosine deaminase (rhADA) or the pan-ADO receptor agonist NECA. Depletion of extracellular ADO with rhADA (*Figure 2J*), or treatment with NECA (*Figure 2K*), also had no impact on mitochondrial respiration or glycolysis of MDA-MB-231 cells. Our results thus indicated that CD73 promoted metabolic fitness independently of ADO signaling.

## CD73 contributes to metabolic fitness of cancer cells through NAD synthesis

NAD + is a coenzyme that plays a central role in mitochondrial respiration (*Verdin, 2015*). We thus investigated whether CD73 could regulate intracellular NAD levels. Using mass-spectrometry (LC-MS), we observed that *NT5E* gene-targeting or antibody-mediated CD73 enzymatic inhibition was associated with a significant reduction of intracellular NAD + levels in MDA-MB-231 (*Figure 3A*) or 4T1.2 cells (*Figure 3—figure supplement 1A*). Depleting extracellular ADO with exogenous rhADA had no effect on intracellular NAD + levels, further supporting an ADO-independent mechanism (*Figure 3A*).

We next tested whether CD73 can serve tumor cells to generate NR, a precursor for NAD synthesis (*Ratajczak et al., 2016*). Using purified recombinant human CD73 or cell surface CD73, in presence or absence of the CD73 inhibitor APCP, we confirmed hydrolysis of exogenous NMN to NR by mass spectrometry (*Figure 2—figure supplement 2B–D*). We then tested the impact of CD73 on the extracellular accumulation of NAD+, NMN and NR from MDA-MB-231 cell cultures. Strikingly, we observed that MDA-MB-231 cell cultures accumulated extracellular NMN and rapidly hydrolyzed it to NR in a CD73-dependent manner (*Figure 3B*). In contrast, CD73 had no impact on extracellular NAD + levels (*Figure 3B*).

Upon intracellular transport, NR must be phosphorylated by intracellular NRK1 to promote intracellular NAD synthesis (*Ratajczak et al., 2016*). We thus tested whether NRK1 was involved

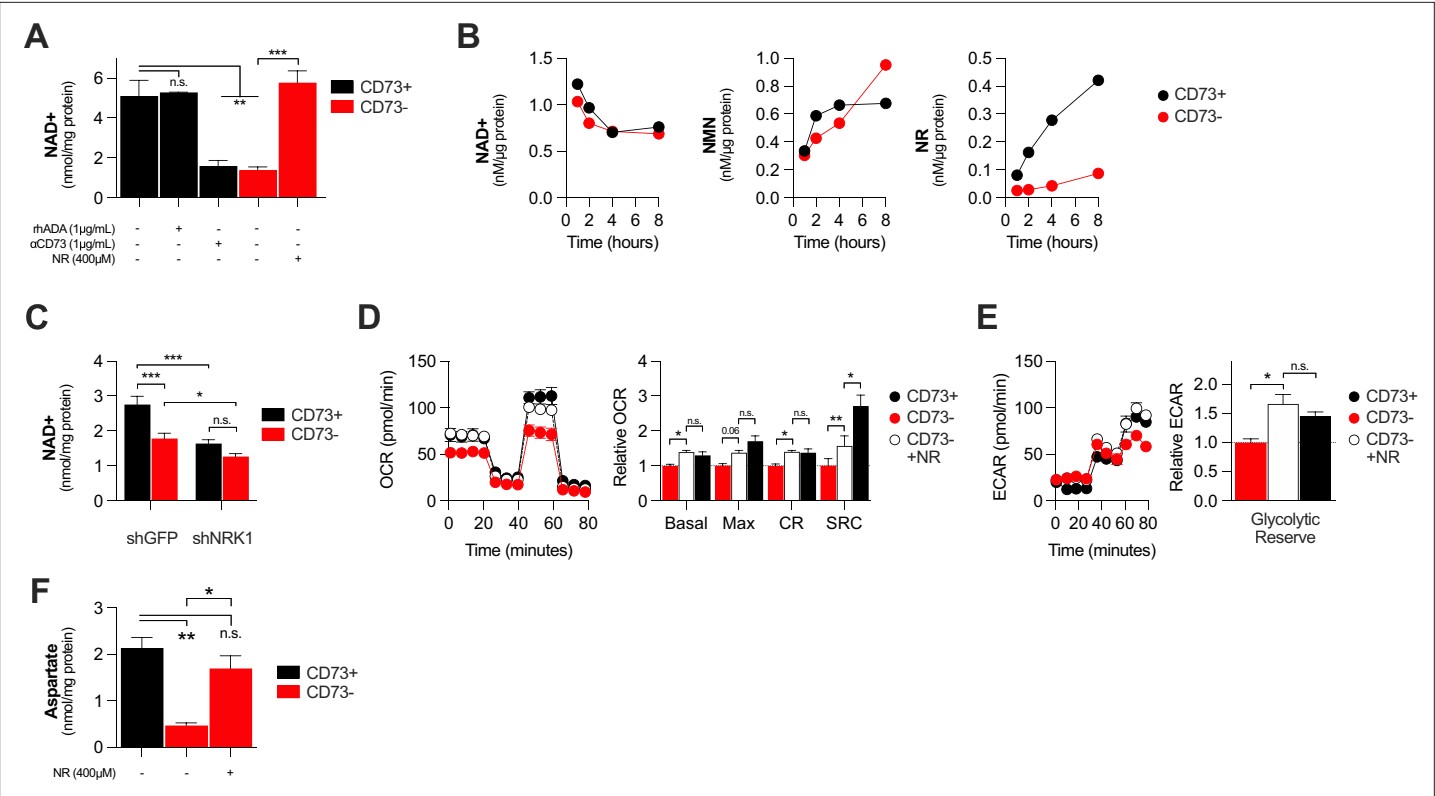

**Figure 3.** CD73 contributes to metabolic fitness of cancer cells through NAD synthesis. (**A**) LC-MS analysis of intracellular nicotinamide adenine levels (NAD+) normalized to protein content of CD73pos (CD73+;+/-1 µg/mL αCD73 mAb or 1 µg/mL rhADA for 48 hr) and CD73neg (CD73-;+/-100 µM NR supplemented 2 x per day for 48 hr) MDA-MB-231 cells (n=2). (**B**) LC-MS analysis of endogenous extracellular production of NAD+ (left panel), NMN (middle panel) and NR (right panel) in supernatant of CD73 +and CD73- MDA-MB-231 cells. Levels were normalized on protein content of adherent cells (n=1). (**C**) LC-MS analysis of intracellular nicotinamide adenine levels (NAD+) normalized to protein content of CD73 + and CD73- MDA-MB-231 cells transfected with either shGFP or shNRK1 (n=2). (**D**) Oxygen consumption rate (OCR) profile (left panel) and OXPHOS parameters (right panel) of CD73- MDA-MB-231 cells cultured with 400 µM NR for 48 hr (n=3). CD73 + cells are shown as a control. (**E**) Extracellular acidification rate (ECAR) profile (left panel) and glycolytic reserve (right panel) of CD73- MDA-MB-231 cells cultured with 400 µM NR for 48 hr (n=3). CD73 + cells are shown as a control. (**F**) Intracellular aspartate levels measured in MDA-MB-231 CD73- cells exposed to exogenous NR (400 µM) for 48 hr. ASP levels were normalized on cell number. CD73 + cells are shown as a control. Means +/- SEM are shown (*p<0.05; **p<0.01; ***p<0.001 by one-way ANOVA).

The online version of this article includes the following figure supplement(s) for figure 3:

**Figure supplement 1.** CD73 contributes to metabolic fitness of cancer cells through NAD synthesis.

in CD73-mediated NAD synthesis. In support of a metabolic role for CD73-mediated hydrolysis of extracellular NMN to NR, shRNA-mediated knockdown of NRK1 (*Figure 2—figure supplement 2E*) prevented CD73 from increasing intracellular NAD + levels in MDA-MB-231 cells (*Figure 3C*). NRK1-deficiency did reduce NAD + levels in both CD73-proficient and CD73-deficient cells, suggesting that some intracellular NR may come from other sources (*Figure 3C*).

To further assess the importance of NR production for CD73 function, we tested whether exogenous NR could rescue the metabolic dysfunctions of CD73neg tumor cells. Indeed, treatment of CD73neg MDA-MB-231 tumor cells with exogenous NR for 48 hr significantly increased their intracellular NAD + levels (*Figure 3A*), their oxygen consumption (*Figure 3D*) and glycolytic reserves (*Figure 3E*), to levels similar to CD73-proficient cultures. Finally, further supporting a role for CD73-NAD axis in mitochondrial metabolism we found that NR supplementation of CD73-deficient cells restored aspartate synthesis (*Figure 3F*). Altogether, our results support a model whereby CD73-dependent extracellular NR promotes intracellular NAD biosynthesis, which in turn enhances mitochondrial respiration and aspartate biosynthesis.

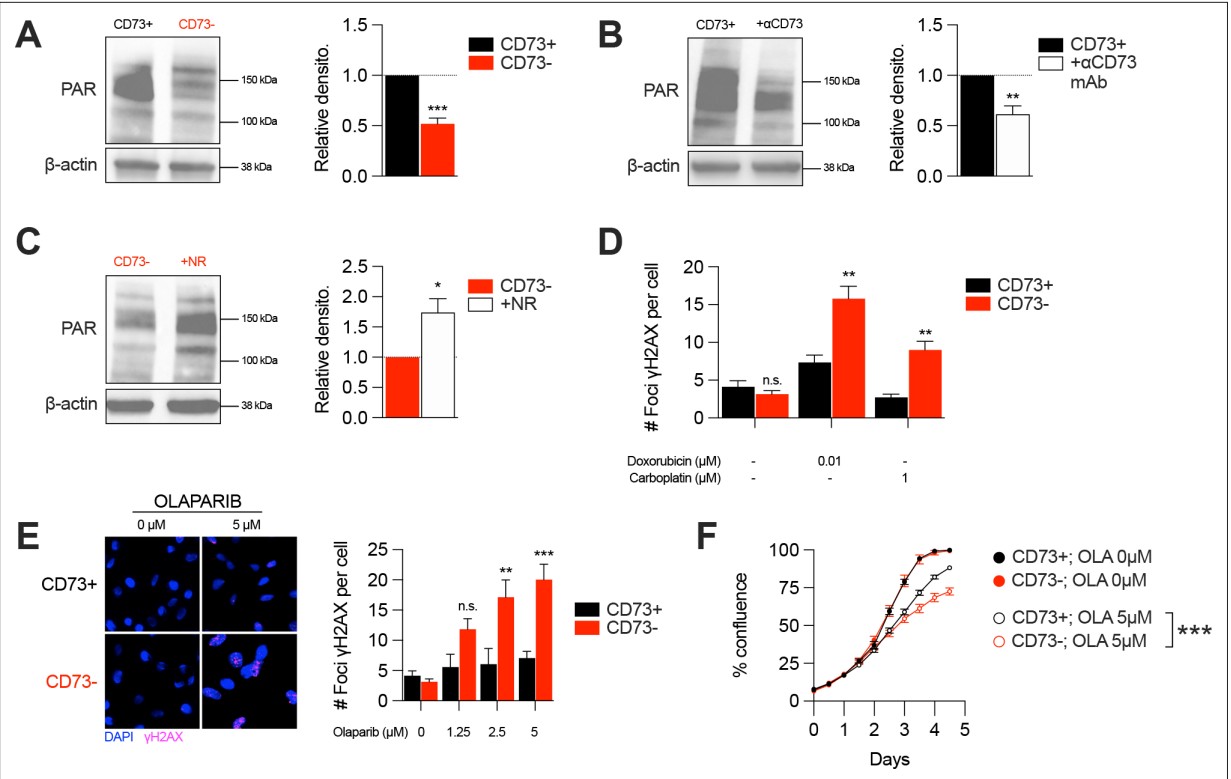

**Figure 4.** CD73 deficiency decreases PARP activity and sensitizes cancer cells to DNA-damaging agents. (**A**) Western blot (left panel) showing PARylation levels in CD73pos (CD73+) and CD73neg (CD73-) MDA-MB-231 cells. Densitometry (right panel) is shown relative to CD73 + cells (n=5). (**B**) Western blot (left panel) showing PARylation levels in CD73 + MDA MB-231 cells treated with αCD73 mAb (1 μg/mL; clone 7G2; 48 hr). Densitometry (right panel) is shown relative to untreated CD73 + cells (n=2). (**C**) Western blot (left panel) showing PARylation levels in CD73- MDA-MB-231 cells treated with NR (400 μM; 48 hr). Densitometry (right panel) is shown relative to untreated CD73- cells (n=2). (**D**) Number of γH2AX foci per CD73 +and CD73- MDA-MB-231 cells treated with doxorubicin (0.01 μM) or carboplatin (1 μM) for 48 hr (n=2). (**E**) γH2AX staining of CD73 + and CD73- MDA-MB-231 cells treated with olaparib (0–5 μM) for 48 hr. Left panels show representative images for γH2AX (magenta) and DAPI (blue) staining. Data show number of γH2AX foci per cells (right panel; n=2). (**F**) In vitro proliferation of CD73 + and CD73- MDA-MB-231 cells treated with (0–5 μM) olaparib for 4 days. Confluence was measured using an Incucyte instrument (n=1). Means +/- SEM are shown (*p<0.05; **p<0.01; ***p<0.001 by Student T tests).

The online version of this article includes the following source data and figure supplement(s) for figure 4:

**Source data 1.** Original raw unedited blots and uncropped blots with the relevant bands labeled that composes *Figure 4A–C*.

**Figure supplement 1.** CD73 deficiency sensitizes UWB1.289+BRCA1 cells to DNA-damaging agents.

**Figure supplement 1—source data 1.** Original raw unedited blots and uncropped blots with the relevant bands labeled that composes *Figure 4— figure supplement 1B*.

## CD73 deficiency decreases PARP activity and sensitizes cancer cells to DNA-damaging agents

In addition of being essential for mitochondrial respiration, NAD is also an important co-factor for PARP enzymes that maintain genomic stability by promoting DNA repair. Interestingly, early studies reported that tumor-derived CD73 can promote chemoresistance (*Loi et al., 2013*; *Ujházy et al., 1996*; *Yu et al., 2021*). To further our understanding of the mechanism of action by which CD73 regulates chemoresistance, we hypothesized that by increasing intracellular NAD levels, CD73 may regulate PARP activity and therefore promote genomic stability. To assess this, we measured PARylation levels by western blotting in $H_2O_2$-treated MDA-MB-231 cells. As shown in *Figure 4*, CD73-deficient tumor cells displayed significantly reduced PARylation compared to CD73-proficient tumor cells (*Figure 4A*). Moreover, targeting CD73 with a neutralizing mAb also reduced PARylation levels, and NR supplementation rescued PARylation in CD73neg cells (*Figure 4B–C*). Similar results were obtained in UWB1.289 human ovarian cancer cells (*Figure 4— figure supplement 1A–B*).

Given the impact of CD73 on PARylation, we next compared the levels of DNA damage in CD73pos and CD73neg tumor cells in response to cytotoxic drugs for TNBC or ovarian cancer. Strikingly, treatment of CD73-deficient MDA-MB-231 or UWB1.289 cells with doxorubicin and carboplatin was associated with significantly greater DNA damage compared to CD73-proficient cells (*Figure 4D*, *Figure 4—figure supplement 1C*). Since decreasing intracellular NAD levels has also been shown to sensitize tumor cells to PARP inhibitors (*Bajrami et al., 2012*), we also evaluated the impact of CD73 on olaparib activity. Accordingly, CD73-deficient MDA-MB-231 cells were significantly more sensitive to olaparib activity in vitro (*Figure 4E–F*). Altogether our results suggest that by regulating intracellular NAD levels, CD73 plays an important role in promoting PARP activity and maintaining genomic stability of tumor cells.

## Discussion

In this study, we demonstrated that the ectonucleotidase CD73, often overexpressed on tumor cells and considered as an important cancer immune checkpoint, enhances tumor cell metabolic fitness in a cell-autonomous manner. CD73 promotes mitochondrial respiration, which generates aspartate required for nucleotide and protein synthesis, and maintains genomic stability notably by increasing the activity of NAD+-dependent PARP enzymes. Surprisingly, these metabolic functions of CD73 were independent of ADO signaling, instead relying on the hydrolysis of extracellular NMN to NR and its metabolism by NRK1.

The importance of CD73 for intracellular NAD synthesis has been controversial (*Wilk et al., 2020*). Using a colorimetric assay, (*Sullivan et al., 2018*) observed no significant effect of inhibiting or deleting CD73 on intracellular NAD concentrations. These discrepancies from our own data may stem from the different sensitivity of the assays (colorimetric versus LC-MS), the different nature of the inhibitors and the different cell lines studied. Notably, Wilk et al. studied MCF-7 cells that express >10-fold lower levels of CD73 compared to the cell lines we analyzed. Importantly, we validated the impact of CD73 on mitochondrial respiration in a large panel of human and mouse tumor cell lines, as well as in primary fibroblasts derived from CD73 gene-targeted mice. Moreover, the fact that NR supplementation restored intracellular NAD +content in CD73-deficient tumor cells, and that CD73 failed to increase NAD +levels in the absence of NRK1 (since NRK1 is necessary and rate-limiting for the generation of NAD +from exogenous NR *Ratajczak et al., 2016*), further support the notion that tumor cell-associated CD73 contributes to maintain intracellular NAD pools.

Intriguingly, we observed that NMN accumulates in the extracellular milieu of proliferating MDA-MB-231 tumor cells. The mechanism leading to such extracellular NMN accumulation remains unclear. We ruled out the culture media as a potential source of NMN. Extracellular NMN may come from NAMPT activity (intracellular or extracellular) on nicotinamide (NAM) that reaches the extracellular milieu as a result of cell death or stress (*Katsyuba et al., 2020*). NAD release may be another source of extracellular NMN via active or passive release mechanisms followed by consumption by glycohydrolases to generate extracellular NAM. Finally, although MDA-MB-231 and 4T1.2 cells do not express CD38, we cannot exclude the involvement of other ecto-NADases.

In the tumor microenvironment, CD38 expression on immune cells may also contribute to consume extracellular NAD+. Production of NAM from NAD-consuming enzymes and NAMPT may thus lead to CD73-mediated NR production. Considering that CD38 can cooperate with CD73 for extracellular ADO production (*Morandi et al., 2018*), it would be of interest to assess the impact of host CD38 on CD73-mediated tumor cell metabolic fitness. Of note, while some studies suggested that CD73 may hydrolyze extracellular NAD+ (*Garavaglia et al., 2012*; *Jablonska et al., 2021*), we and others (*Wilk et al., 2020*) did not observe this.

Consistent with the observed decrease in intracellular NAD pools, and the importance of NAD for glycolysis (*Hopp, 2019*; *Luengo et al., 2021*), we found that CD73-deficient tumor cells also displayed impaired maximal glycolytic capacity following oligomycin treatment, which inhibits ATP synthase and forces cells to redirect pyruvate to lactate conversion via glycolysis (*Luengo et al., 2021*).

We also demonstrated that CD73-deficient tumor cells had significantly reduced PARP activity, which rely on NAD + as cofactor. Accordingly, targeting CD73 sensitized tumor cells to chemotherapy and PARP inhibition. Other DNA repairing enzymes besides PARPs also require NAD+ (*Ruszkiewicz et al., 2022*), including the sirtuins SIRT1 and SIRT6 that play critical roles in promoting homologous recombination (HR) and nonhomologous end joining (NHEJ) repairs. It would therefore be of interest

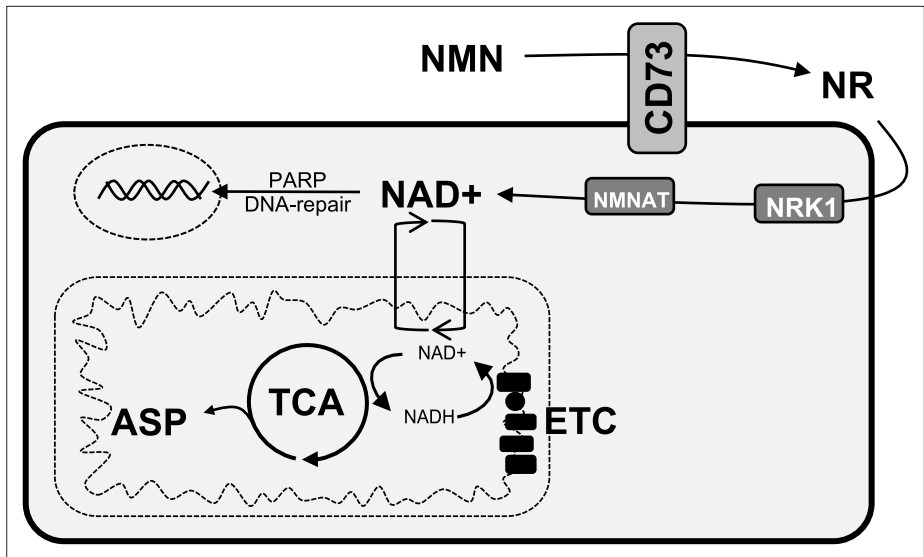

**Figure 5.** Schematic representation of the role of CD73 in NAD + biosynthesis, metabolic function, and genomic stability in cancer cells. We propose that CD73 regulates to NAD + biosynthesis intracellularly in a NRK1-dependent manner by hydrolyzing extracellular NMN into NR, which in turn favors DNA-damage response, mitochondrial respiration, aspartate synthesis and tumor growth intrinsically. NMN = nicotinamide mononucleotide, NR = nicotinamide riboside, ASP = aspartate, ETC = electron transport chain.

to further evaluate the impact of CD73 on sirtuin-mediated HR and NHEJ. A broad impact of CD73 on DNA repair mechanisms might explain the observation that CD73-deficient tumor cells become sensitive to PARP inhibition despite expressing BRCA. Other studies have shown that targeting NAD +synthesis (by blocking NAMPT) enhances the activity of olaparib in BRCA1-competent TNBC cells (*Bajrami et al., 2012*). Alternatively, targeting NAD + synthesis may decrease BRCA1 expression or activity, as was previously reported (*Li et al., 2014*).

Interestingly, CD73 deficiency in mice was found associated with reduced serum arginine levels (*Mierzejewska et al., 2019*). In addition of diet, another source of arginine is through the metabolism of intracellular aspartate via cytosolic argininosuccinate synthase 1 (ASS1) (*Garcia-Bermudez et al., 2020*). Our study highlighted the importance of CD73 in promoting intracellular aspartate synthesis. It is therefore possible that the systemic reduction in arginine levels observed in CD73-deficient mice may stem from dysfunctional aspartate biosynthesis.

In conclusion, our study sheds new light on the immune-independent functions of CD73 (*Figure 5*). We demonstrated that CD73 promote tumor growth in a cancer-cell autonomous manner by increasing metabolic fitness. This in turn favors cancer cell adaptation to nutrient-deprived micro-environments and cytotoxic stress by increasing OXPHOS, aspartate biosynthesis and DNA repair. Our study highlights new therapeutic opportunities that may be exploited in novel combination anti-cancer strategies.

## Methods

### Key resources table

| Reagent type (species) or resource | Designation | Source or reference | Identifiers | Additional information |
|---|---|---|---|---|
| Antibody | Anti-human CD73 (7G2, mouse monoclonal) | Invitrogen | Cat#410200, RRID: AB_2533492 | Inhibition (1 µg/mL) |
| Antibody | Anti-human CD73 BV421 (mouse monoclonal) | BD Bioscience | Cat#562430, RRID: AB_11153119 | FACS (1:200) |
| Antibody | Anti-mouse CD73 PE-Cy7 (Rat monoclonal) | Thermo Fisher Scientific | Cat#25-0731-82, RRID: AB_10853348 | FACS (1:200) |

*Continued on next page*

*Continued*

| Reagent type (species) or resource | Designation | Source or reference | Identifiers | Additional information |
|---|---|---|---|---|
| Antibody | Anti-β-actin (mouse monoclonal) | Abcam | Cat#ab8226 | WB (1:5000) |
| Antibody | Anti-PLCγ (mouse polyclonal) | Millipore | Cat#05–163 | WB (1:2000) |
| Antibody | Anti-human CD73 (mouse monoclonal) | Abcam | Cat#Ab91086 | WB (1:1000) |
| Antibody | Anti-FLAG (rabbit polyclonal) | Millipore | Cat#F7425 | WB (1:1000) |
| Antibody | Anti-Cas9 *S. pyogenes* (rabbit monoclonal) | Cell Signaling Technology | Cat#19526 S | WB (1:1000) |
| Antibody | Anti-PAR (mouse monoclonal) | Trevigen | Cat#4335-MC-100 | WB (1:1000) |
| Antibody | HRP-conjugated anti-mouse IgG secondary (goat polyclonal) | Millipore | Cat#AP124P | WB (2.5:10,000) |
| Antibody | IRDye 800CW anti-Mouse IgG secondary (donkey polyclonal) | LI-COR | Cat#926–32212, RRID: AB_621847 | WB (1:10,000) |
| Antibody | IRDye 680RD anti-rabbit IgG secondary (donkey polyclonal) | LI-COR | Cat#926–68073, RRID: AB_10954442 | WB (1:10,000) |
| Antibody | Anti-phospho-Histone H2A.X Ser139 (mouse monoclonal) | Millipore | Cat#05–636 | IF (1:2000) |
| Antibody | Anti-Mouse IgG (H+L) Highly Cross-Adsorbed Secondary Alexa Fluor 488 (donkey polyclonal) | Thermo Fisher Scientific | Cat#A-21202, RRID: AB_141607 | IF (1:800) |
| Chemical compound, drug | Adenosine 5'-(α,β-methylene) diphosphate (APCP) | Sigma-Aldrich | Cat#M3763, CAS: 3768-14-7 | |
| Chemical compound, drug | Nicotinamide riboside (NR) | Cayman Chemical | Cat#23132, CAS: 1341-23-7 | |
| Chemical compound, drug | Nicotinamide mononucleotide (NMN) | Sigma-Aldrich | Cat#N3501, CAS: 1094-61-7 | |
| Chemical compound, drug | 5'-N-Ethylcarboxamidoadenosine (NECA) | Tocris | Cat#1691, CAS: 35920-39-9 | |
| Chemical compound, drug | SCH58261 | Tocris | Cat#2270, CAS: 160098-96-4 | |
| Chemical compound, drug | PSB1115 | Tocris | Cat#2009, CAS: 152529-79-8 | |
| Chemical compound, drug | Olaparib (AZD2281) | Selleckchem | Cat#S1060, CAS: 763113-22-0 | |
| Chemical compound, drug | Doxorubicin | CRCHUM pharmacy | CAS: 23214-92-8 | |
| Chemical compound, drug | Carboplatin | CRCHUM pharmacy | CAS: 41575-94-4 | |
| Chemical compound, drug | 2-deoxy-2-[(7-nitro-2,1,3-benzoxadiazol-4-yl)amino]-D-glucose (2-NBDG) | Cayman Chemical | Cat#11046, CAS: 186689-07-6 | |
| Chemical compound, drug | U-[$^{13}$C]-glucose | Cambridge Isotope Laboratories | Cat#CLM-1396, CAS: 110187-42-3 | |
| Chemical compound, drug | U-[$^{13}$C]-glutamine | Cambridge Isotope Laboratories | Cat#CLM-1822, CAS: 184161-19-1 | |
| Chemical compound, drug | Oligomycin | Sigma-Aldrich | Cat#O4876, CAS: 1404-19-9 | |
| Chemical compound, drug | Carbonyl cyanide p-trifluoromethoxyphenylhydrazone (FCCP) | Sigma-Aldrich | Cat#C2920, CAS: 370-86-5 | |

*Continued on next page*

*Continued*

| Reagent type (species) or resource | Designation | Source or reference | Identifiers | Additional information |
|---|---|---|---|---|
| Chemical compound, drug | Rotenone | Sigma-Aldrich | Cat#R8875, CAS: 83-79-4 | |
| Chemical compound, drug | Antimycin A from *Streptomyces* sp. | Sigma-Aldrich | Cat#A8674, CAS: 1397-94-0 | |
| Chemical compound, drug | CelLytic M buffer | Sigma-Aldrich | Cat#C2978 | |
| Chemical compound, drug | Halt protease and phosphatase cocktail inhibitors | Sigma-Aldrich | Cat#78440 | |
| Chemical compound, drug | Bradford protein assay dye reagent | Bio-Rad | Cat#500–0006 | |
| Chemical compound, drug | Protein Block DAKO solution | Agilent | Cat#X0909 | |
| Chemical compound, drug | Fluoromount Aqueous Mounting Medium | Sigma-Aldrich | Cat#F4680 | |
| Chemical compound, drug | Formalin solution, neutral buffered, 10% | Sigma-Aldrich | Cat#HT5011 | |
| Chemical compound, drug | Triton X-100 reduced | Sigma-Aldrich | Cat#X100RS; CAS: 92046-34-9 | |
| Chemical compound, drug | 7-amino-actinomycin D (7-AAD) | eBioscience | Cat#00-6993-50 | |
| Peptide, recombinant protein | Recombinant human adenosine deaminase (rhADA) | R&D systems | Cat#7048-AD, Acc#P00813 | |
| Peptide, recombinant protein | Recombinant human CD73 (rhCD73) | R&D systems | Cat#5795-EN, Acc#AAH659937 | |
| Commercial assay, kit | Malachite green phosphate detection kit | R&D systems | Cat#DY996 | |
| Commercial assay, kit | SuperScript VILO cDNA Synthesis Kit | Thermo Fisher Scientific | Cat#11754050 | |
| Commercial assay, kit | RNeasy mini kit | QIAGEN | Cat#74104 | |
| Commercial assay, kit | ALDEFLUOR kit | Stemcell | Cat#01700 | |
| Commercial assay, kit | Amersham ECL prime detection reagent | GE healthcare lifescience | Cat#RPN2232 | |
| Commercial assay, kit | Glutamine/Glutamate-Glo Assay kit | Promega | Cat#J8021 | |
| Cell line (*homo-sapiens*) | MDA-MB-231 (TNBC) | ATCC | HTB-26 | |
| Cell line (*homo-sapiens*) | PANC-1 (pancreatic cancer) | ATCC | CRL-1469 | |
| Cell line (*homo-sapiens*) | SK23MEL (metastatic melanoma) | MSK Cancer Center | RRID: CVCL_6027 | |
| Cell line (*homo-sapiens*) | HCC1954 (breast cancer) | ATCC | CRL-2338 | |
| Cell line (*homo-sapiens*) | OV1369 (ovarian cancer) | Dr. Anne-Marie Mes-Masson | | *Fleury et al., 2019* |
| Cell line (*homo-sapiens*) | SKOV3 (ovarian cancer) | ATCC | HTB-77 | |

*Continued on next page*

*Continued*

| Reagent type (species) or resource | Designation | Source or reference | Identifiers | Additional information |
|---|---|---|---|---|
| Cell line (*homo-sapiens*) | HEK293FT (human embryonic kidney) | Dr. Francis Rodier | RRID: CVCL_6911 | |
| Cell line (*homo-sapiens*) | UWB1.289+BRCA1 | Dr. Madhuri Koti | RRID: CVCL_B078 | |
| Cell line (*Mus musculus*) | 4T1.2 (mammary cancer) | ATCC | CRL-2539 | |
| Cell line (*Mus musculus*) | SM1 LWT1 (metastatic melanoma) | Dr. Mark Smyth | | |
| Cell line (*Mus musculus*) | MEF from C57BL/6 J | This paper | | Generated in Dr. Stagg's laboratory |
| Cell line (*Mus musculus*) | MEF from *Nt5e*-/- C57BL/6 J | This paper | | Generated in Dr. Stagg's laboratory |
| Strain, strain background (mouse) | NOD.Cg-Rag1tm1Mom Il2rgtm1Wjl/SzJ (NRG) | Jackson Laboratory | JAX: 007799 | |
| Strain, strain background (mouse) | C57BL/6 J | Jackson Laboratory | JAX: 000664 | |
| Strain, strain background (mouse) | B6.129S1-Nt5etm1Lft/J (CD73-/- C57BL/6 J) | Dr. Linda Thompson | JAX: 018986 | |
| Sequence-based reagent | *Nt5e* | Thermo Fisher Scientific | Cat#4331182, Mm00501910_m1 | |
| Sequence-based reagent | *Actb* | Thermo Fisher Scientific | Cat#4331182, Mm00607939_s1 | |
| Sequence-based reagent | *NMRK1* | Thermo Fisher Scientific | Cat#4331182, Hs00944470_m1 | |
| Sequence-based reagent | *ACTB* | Thermo Fisher Scientific | Cat#4331182, Hs99999903_m1 | |
| Sequence-based reagent | *ADORA2A* | Thermo Fisher Scientific | Cat#4331182, Hs00169123_m1 | |
| Sequence-based reagent | *ADORA2B* | Thermo Fisher Scientific | Cat#4331182, Hs00386497_m1 | |
| Recombinant DNA reagent | pLenti-PGK-ER-KRAS G12V (plasmid) | Dr. Francis Rodier | | *Rodier et al., 2009*; Addgene#35635 |
| Recombinant DNA reagent | pLHCX (plasmid) | Dr. Lucas Sullivan | | *Sullivan et al., 2018* |
| Recombinant DNA reagent | pLHCX-gpASNase1 | Dr. Lucas Sullivan | | *Sullivan et al., 2018*; Addgene#121526 |
| Recombinant DNA reagent | shGFP | Sigma-Aldrich | TRCN0000075219 | |
| Recombinant DNA reagent | shNRK1 | Sigma-Aldrich | TRCN0000160469 | |
| Other | DMEM | Wisent | | Cell culture reagent |
| Other | Glucose/glutamine/sodium pyruvate-free DMEM | Wisent | | Cell culture reagent |
| Other | RPMI 1640 | Wisent | | Cell culture reagent |
| Other | OSE | Wisent | | Cell culture reagent |
| Other | Dialyzed FBS | Wisent | | Cell culture reagent |
| Other | Seahorse base media | Agilent | | Cell culture reagent |

*Continued*

| Reagent type (species) or resource | Designation | Source or reference | Identifiers | Additional information |
|---|---|---|---|---|
| Other | Glutamax | Gibco | Cat#35050–061 | Cell culture reagent |
| Other | D-(+)-glucose (cell culture grade) | Sigma-Aldrich | Cat#G7021, CAS: 50-99-7 | Cell culture reagent |
| Other | L-glutamine (cell culture grade) | Sigma-Aldrich | Cat#G8540, CAS: 56-85-9 | Cell culture reagent |
| Software, algorithm | FlowJo (10.8.0) | FlowJo | RRID:SCR_008520 | |
| Software, algorithm | GraphPad Prism 9 (9.2.0) | GraphPad | RRID:SCR_002798 | |
| Software, algorithm | StepOne (2.3) | Thermo Fisher Scientific | RRID:SCR_014281 | |
| Software, algorithm | Incucyte base analysis software | Sartorius | RRID:SCR_019874 | |
| Software, algorithm | Seahorse Wave software | Agilent | RRID:SCR_014526 | |
| Software, algorithm | Image Lab Software (6.0.1) | Bio-Rad | RRID:SCR_014210 | |
| Software, algorithm | Visiomorph software | B&B microscopes | http://www.bbmicro.com/products/product.php?pid=204 | |

## Cell lines generation and culture

*NT5E* gene-editing was generated by electroporation of all-in-one CRISPR/Cas9 vector (px330, Addgene) expressing the 20mer target sequence GCAGCACGTTGGGTTCGGCG (exon1), provided by Michael Hoelzel (University of Bonn, Germany). Cells were sorted based on CD73 expression (CD73 +and CD73-) within 1 week after transfection. Enzymatic functionality was verified using the commercially available malachite green kit assay (R&D system).

Empty vector (pLHCX-ev) and plasmid coding for a FLAG-tagged guinea pig asparaginase (pLHCX-gpASNase1) enzyme were kindly provided by Dr. Lucas B. Sullivan (Fred Hutchinson Cancer Research Center, Seattle, WA, USA; *Sullivan et al., 2018*). CD73 +and CD73- MDA-MB-231 cells were infected with a retrovirus containing either pLHCX-ev or pLHCX-gpASNase1 and selected with 750 µg/mL of Hygromycin B-containing media until all uninfected control cells died. For NRK1 knockdown in MDA-231 cells, CD73 +and CD73- cells were infected with a retrovirus containing either a control shRNA (shGFP) or shNRK1 (Sigma) and selected with 1 µg/mL of puromycin-containing media until all uninfected control cells died. Plasmids were purchased from Sigma.

MEF were isolated from embryos at 14.5 days of gestation from WT and *Nt5e*$^{-/-}$ C57BL/6 mice. Briefly, pregnant females were euthanized, uterus were dissected out and individual embryos separated from their yolk sacs into PBS-containing petri dishes. Limbs, tail, head and liver were cut off embryos before being digested and broken up by up-and-down pipetting in freshly thawed 0.25% trypsin solution and incubated at 37 °C in water bath for 5 min cycles until only insoluble cartilage would remain and not settle down. Pooled dissociated cells were filtered through a 40 µM mesh, centrifuged 10 min at 1200 rpm, resuspended in DMEM with 1% pen/strep antibiotics, seeded in 100 mm x 15 mm petri dishes (2 plates per embryo) and incubated in a 37 °C/5% $CO_2$ incubator. Growth media was replaced the next day and rapidly transformed within 7 days upon isolation with a lenti-viral vector expressing an oncogenic K-Ras$^{G12V}$ (pLenti PGK RasV12), kindly provided by Dr. Francis Rodier (Université de Montréal, Montréal, QC, CA; *Rodier et al., 2009*). Lentivirus was generated in HEK293FT cells by transfection using standard techniques and cells selected with 75 µg/mL of hygromycin B until all uninfected cells were dead. Immortalized MEF clones upon K-Ras transformation were pooled and used for experimentation.

Cells were purchased from ATCC and frozen aliquots were thawed and used for study within 20 in vitro passages without re-authentication. All cells were grown at 37 °C in a 5% $CO_2$ humid atmosphere. Cells were sub-cultured or media was changed every 3 days. Cell lines were routinely tested and confirmed negative for mycoplasma. MDA-MB-231, MDA-MB-231 ALDH$^{hi}$, 4T1.2, SM1 LWT1, PANC-1 cell lines were cultured in antibiotics-free DMEM (Wisent) supplemented with 10% FBS (referred to as growth media). SK23MEL and HCC1954 cell lines were cultured in antibiotics-free RPMI 1640 (Wisent)

supplemented with 10% FBS. SKOV3 cells were cultured in OSE (Wisent) supplemented with 10% FBS and 1% pen/strep antibiotics. Mouse embryonic fibroblasts (MEF; generated and immortalized as described in method details) were cultured in DMEM (Wisent) supplemented with 10% FBS and 1% pen/strep antibiotics. UWB1.289+BRCA1 cells were culture in 50/50 RPMI (Wisent) and MEGM (Lonza) media supplemented with 3% FBS and 200 µg/mL G-418. All cell lines were negative for mycoplasma (Lonza, MycoAlert). Pyruvate-, glucose- and glutamine-free DMEM media (Wisent) is defined as assay media and supplementations of the media are described in method details depending on the experiment.

## Animal experimentation

NOD-*Rag1*[null] *IL2rg*[null] (NRG; Jackson Laboratory) and *Nt5e*[-/-] C57BL/6 J mice (obtained from Dr. Linda Thompson; OMRF, Oklahoma City, OK, USA) were bred and housed at the CRCHUM which holds a certificate of Good Animal Practice from the Canadian Council on Animal Care (CCAC). C57BL/6 J mice were purchased from Jackson Laboratory and housed at the CRCHUM. All the experimental procedures were authorized, and all animals were handled according to an approved Institutional Animal Care and Use Committee (IACUC) protocol (#C20010JSs) of the Centre de Recherche du Centre Hospitalier de l'Université de Montréal. For in vivo tumor growth, 8–12 weeks-old healthy female NRG mice were injected with $2 \times 10^6$ MDA-MB-231 cells sub-cutaneously and tumor growth was monitored thrice weekly. Tumor volume was measured by caliper in two dimensions, and volumes were estimated using the equation V = (large diameter ×small diameter$^2$)×0.5. The total number of mice analyzed for each experiment is detailed in figure legends. C57BL/6 J and *Nt5e*[-/-] C57BL/6 J mice were used for mouse embryoblastic fibroblast (MEF) derivation (see cell lines generation and culture section).

## Flow cytometry

Single cell suspensions were strained (40 µM nylon mesh) and incubated with fluorescence-conjugated antibodies (1:200 PE-Cy7 rat anti-mouse CD73 or 1:200 BV421 mouse anti-human CD73) for 30 min at 4 °C in ice-cold PBS + 2% FBS+2 mM EDTA (FACS buffer). Fluorescence was acquired with a LSRFortessa flow cytometer (BD) or sorted with a FACSAria III cell sorting system (BD) equipped with FACSDiva software (BD). Dead cells were excluded using a fixable viability dye eF506 (Thermo Fisher Scientific; cat.: 65-0866-14). FACS data were further analyzed with FlowJo software (version 10.8.0). For viability assay, cells were stained with 7-AAD (1:20) for 15 min at room temperature, washed twice with FACS buffer and immediately acquired on a FACS machine.

## Glucose incorporation assay

Cells were cultured in assay media supplemented with 10% FBS and 2 mM glutamine (Sigma) for 30 min before being harvested, washed with PBS 1 X twice and incubated with 100 µM of 2-NBDG (Cayman Chemical) for 30 min at 37 °C in a 5% $CO_2$ humid atmosphere. Cells were washed twice with PBS 1 X and resuspended in FACS buffer (see flow cytometry section). Fluorescence was analyzed in the FITC channel on a LSRFortessa flow cytometer (BD).

## Glutamine depletion assay

10 000 MDA-MB-231 CD73 +and CD73- cells were seeded per wells of a 96-well plate and cultured in assay media complemented with 2 mM glutamine (Sigma), 25 mM glucose (Sigma) and 10% FBS for 3 days. Supernatant was collected daily and stored at –20 °C until analysis. Glutamine levels in media was measured using a commercially available kit (Promega) according to manufacturer's protocol.

## ALDH assay

Tumor-initiating cells were sorted based on ALDH activity and tested on seahorse analyzer as previously described (*Lee et al., 2017*) using the ALDEFLUOR kit (Stemcell). Briefly, cells were harvested and resuspended at $2.5 \times 10^6$ cells/mL of ALDEFLUOR assay buffer. Five µL/mL of the ADLEFLUOR reagent was added to each cell line test tubes and 0.5 mL per test tubes were immediately aliquoted in control tubes in which 10 µL of ALDEFLUOR DEAB reagent were added. Cells were incubated for 30 min at 37 °C washed and resuspended in ALDEFLUOR assay buffer and rapidly sorted on a FACSAria III cell sorting system (BD). $5 \times 10^4$ freshly sorted ALDH[hi] cells were seeded in seahorse XF24

plate in growth media and incubated overnight before seahorse analysis as described in analysis of extracellular flux section.

## Proliferation assays

A total of $2.5 \times 10^3$ cells were seeded in a 48-well plate and cultured in growth media for 8 hr before being washed and switched to assay media supplemented with 10% FBS and 0.1–2 mM glutamine (Glutamax; Gibco) and/or 2.5–25 mM glucose (Sigma). The moment at which the media was changed is defined as baseline (day 0). Percentage of well confluency was measured by applying a phase mask over 4 X photos (2 photos per well) taken every 2 hr for 5 days using the Incucyte technology. For each independent experiments, experimental conditions were tested in triplicate. Proliferation was calculated by reporting the confluence level every 0.5 days relative to confluence level at baseline for each well.

## Stable isotope tracing analysis (SITA)

For SITA, $3 \times 10^6$ cells/well were seeded in a six-well plate and cultured in 2 mL assay media supplemented with 10% dialyzed FBS (Wisent) and either 25 mM U-[$^{13}$C]-glucose (Cambridge Isotope Laboratories) and 4 mM glutamine (Sigma) or 4 mM U-[$^{13}$C]-glutamine (Cambridge Isotope Laboratories) and 25 mM glucose (Sigma) for 3 hr at 37 °C. Metabolites were extracted from cells using dry ice-cold 80% methanol, followed by sonication and removal of cellular debris by centrifugation at 4 °C. Metabolite extracts were dried, derivatized as tert-butyldimethylsilyl esters, and analyzed via GC-MS by the metabolomic core from the *Goodman cancer research centre* (McGill University). Labeled metabolite abundance was expressed relative to the internal standard D27 (D-myristic acid) and normalized to protein content. Mass isotopomer distribution was determined using a custom algorithm developed at McGill University (*Ma et al., 2017*).

## LC-MS analyses

For intracellular measurements of AMP, ADP, ATP, NAD + and aspartate, $3 \times 10^6$ cells were seeded in 100 mm petri dishes. Cells were incubated with drugs (7G2 mAb, rhADA or NR) for 48 hr in assay media supplemented with 25 mM glucose, 2 mM Glutamax and 10% dialyzed FBS. Media was washed once with PBS 1 X, remove completely and cell culture dishes were quickly snap frozen on liquid nitrogen. Cells were scraped on ice and collected in 675 µL ice-cold extraction buffer [80% (vol/vol) methanol, 2 mM ammonium acetate, pH 9.0, with 10 µM AMP-13C10,15N5 (IS-AMP) as internal standard], transferred into polypropylene (PP) tubes, and sonicated in a cup-horn sonicator at 150 W for 2 min (cycles of 10 s on, 10 s off) in an ethanol/ice bath. Cell extracts were centrifuged at 4 °C for 10 min at 25,830×*g*, and supernatants were collected in ice-cold 2 mL PP tubes, to which 250 µL water was added. Polar metabolites were extracted with 1080 µL of chloroform:heptane (3:1 vol/vol) by 2×10 s vortex followed by 10 min incubation on ice and 15 min centrifugation at 4 °C, 12,500×*g*. From the upper phase, 600 µL was collected without carrying out any interface material and transferred into new cold 2 mL PP tubes. These tubes were centrifuged again, and 400 µL supernatant were collected into cold 1.5 mL PP tubes. Samples were frozen in liquid nitrogen and dried in two steps – first, in a SpeedVac Concentrator for ~2 hr (maximal vacuum, no heat; Savant) at 4 °C to remove methanol; and second, by lyophilization for 90 min (Labconco FreeZone) – and then stored at –80 °C until used. Samples were reconstituted in 14 µL of Milli-Q water, and injections of 3 µL were performed in duplicate on an electrospray ionization LC-MS/MS system composed of an Agilent 1200 SL device (for LC) and a triple-quadrupole mass spectrometer (4000Q TRAP MS/MS; Sciex). Samples were separated by gradient elution of 12 min on a Poroshell 120 EC-C18, 2.1×75 mm, 2.7 µm column (Agilent Technologies) using mobile phase consisting of an aqueous solvent A (10 mM tributylamine, 15 mM acetic acid, pH 5.2) and an organic solvent B (95% (vol/vol) acetonitrile in water, 0.1% formic acid), at a flow rate of 0.75 mL/min and column oven temperature of 40 °C. Quantification was performed as described in *Guay et al., 2013*.

For extracellular measurements of NAD+, NMN and NR, 0.2 µg/mL of human recombinant CD73 or $5 \times 10^4$ cells seeded in a 96-wells plate were cultured in assay buffer (2 mM MgCl$_2$, 125 mM NaCl, 1 mM KCl, 10 mM Glucose, 10 mM HEPES pH 7.2, ddH$_2$O) with or without exogenous NMN (0.2 mM) for 1–8 hr. Supernatant was collected in 40% methanol +40% acetonitrile dry ice-cold extraction solution (2:2:1 methanol:acetonitrile:aqueous ratio). Samples were flash frozen on dry ice and stored at

–80 °C. For LC-MS/MS analysis, samples (500 µL) were thawed and AMP-13C10,15N5 (IS-AMP) was added as internal standard. Samples were incubated 1 hr at 4 °C with vigorous agitation and centrifuged 10 min, 20K×g, 4 °C. Supernatants were transferred to new polypropylene tubes, dried down at 10 °C by centrifugal vacuum evaporation, then reconstituted in 50 µL 80% acetonitrile in water before LC-MS/MS analysis. Samples (5 µL injections) were separated by HILIC (Nexera X2, Shimadzu) at 0.25 mL/min using a gradient elution (A=MilliQ water, B=90% acetonitrile in water, both A and B containing 10 mM ammonium acetate pH 9.0 and 5 µM Agilent's deactivator additive; gradient: 0 min 90% B, 2 min 90% B, 12 min 60% B, 15 min 60% B, 16 min 90% B (re-equilibration), 24 min 90% B) on a Poroshell 120 HILIC-Z 2.1×100 mm, 2.7 µm HPLC column (Agilent) following a guard column of similar material, both kept at 30 °C. Metabolites were detected after ESI on a triple quadrupole mass spectrometer (QTRAP 6500, SCIEX) with polarity switching looking for the following transitions: negative ions IS-AMP 360.8/79.0, NAD 661.8/540.0; positive ions NR 255.0/123.1, NMN 334.9/123.2. Depending on the experiment, results normalized on protein content are shown as area of peak obtain by mass spectrometry analysis or absolute concentrations. For quantification of absolute concentration, a mix of pure standards prepared in 80% acetonitrile in water was used to build a calibration curve. No traces of NAD, NMN or NR were found in the assay buffer.

## Extracellular flux analysis

Oxygen consumption rate (OCR) and extracellular acidification rate (ECAR) were measured in real-time using Seahorse (XF24 and XFe96) extracellular flux analyzers (Agilent). Cells were seeded in XF24 ($5 \times 10^4$) or XFe96 ($1 \times 10^4$) and incubated between 12- and 48 hr depending on the experiment. For assays with 7G2 mAb, SCH58261, PSB1115, rhADA, NECA or ADO, drugs were added to the media after seeding and changed after 24 hr with fresh media. The day of the assay, cells were washed and incubated for 60 min in a $CO_2$-free incubator in seahorse assay base media (Agilent) containing 25 mM glucose (Sigma) and 2 mM glutamine (Sigma). For mitochondrial stress test, ATP synthase was inhibited by injection of 1 µM oligomycin (Sigma), followed by 1 µM FCCP (Sigma) to induce mitochondrial uncoupling to determine the spare/maximal respiratory capacity. Non-mitochondrial respiration was determined after rotenone (Sigma; 1 µM)/antimycin A (Sigma; 0.1 µM) injection. For glycolysis measurements, cells were glucose-starved for 60 min in seahorse assay base media containing 2 mM glutamine (Sigma). Glycolysis was measured upon glucose injection (Sigma; 25 mM) and glycolytic reserve upon oligomycin (Sigma; 1 µM) injection. Upon completion of the assay, data were normalized by cell counts (XF24) or crystal violet (XFe96) to correct for seeding density and analysis was performed using Seahorse Wave software (2 min mix, 2 min wait, 2 min measurements).

## Genome-wide transcriptional analysis

Total RNA was isolated from MDA-MB-231 cells using the RNeasy Plus Mini Kit (Qiagen) according to the manufacturer's instructions. RNA was quantified using a DS-11 spectrophotometer (DeNovix). Samples were sent to *Génome Québec* (Montréal, QC, CA) for gene expression analysis using an Affymetrix microarray chip. Expression values were computed using the robust multi-array analysis (RMA) normalization method ('affy' package in Bioconductor; *Irizarry et al., 2003*; *Gautier et al., 2004*). When multiple probe sets mapped to the same official gene symbol, we computed their average value. Differential expression (DE) analysis was performed using 'limma' (*Ritchie et al., 2015*) standard analysis pipeline. Significant DE gene were defined as genes with absolute fold-change ≥1.5 and adjusted p-value ≤0.05. Statistical analyses were performed between *NT5E* mRNA expression and genes expression using Spearman correlation.

## Quantitative real-time PCR analysis

Total RNA was isolated and quantified as described in genome-wide transcriptional analysis section. Reverse transcription of 1 µg RNA was performed using the SuperScript VILO cDNA Synthesis Kit (Thermo Fisher Scientific) and *ADORA2A*, *ADORA2B*, *NMRK1* and *Nt5e* gene expression encoding for A2A, A2B, NRK1 and CD73 respectively was analyzed using Taqman probes for mouse *Nt5e* (Thermo Fisher Scientific; Mm00501910_m1), or human *NMRK1* (Thermo Fisher Scientific; Hs00944470_m1), human *ADORA2A* (Thermo Fisher Scientific; Hs00169123_m1) or human *ADORA2B* (Thermo Fisher Scientific; Hs00386497_m1) relative to mouse *Actb* (Thermo Fisher Scientific; Mm00607939_s1) or

human *ACTB* (Thermo Fisher Scientific; Hs99999903_m1) using the StepOne PCR machine (Thermo Fisher Scientific) and software (version 2.3).

## Western blot

Adherent cells were washed with ice-cold PBS and lysed and scrapped in CelLytic M buffer (Sigma) with 1 X Halt protease and phosphatase cocktail inhibitors (Thermo Fisher Scientific) before being centrifuged for 15 min at 20,000 $g$ at 4 °C. Proteins were harvested from supernatant and quantified by using Bradford protein assay dye reagent (Bio-Rad). Twenty-five µg of protein from whole cell extract were loaded in 4–10% acrylamide gels and transferred on nitrocellulose membranes. Membranes were stained overnight in 5% BSA-containing PBS Tween 0.1% with following antibodies: mouse anti-β-actin (1:5000), mouse anti-PLCγ (1:2000), mouse anti-CD73 (human; 1:1000), mouse anti-FLAG (1:1000) and rabbit anti-Cas9 (*S. pyogenes*; 1:1000). For PARylation assay, $1 \times 10^6$ cells were seeded in six-well plates and drugs was added in media for 48 hr. Before proceeding to protein extraction, cells were treated with 2 mM $H_2O_2$ for 20 min at room temperature. A total of 50 µg proteins were load in a 4–10% acrylamide gels, transferred on nitrocellulose membrane and stained overnight in 5% milk-containing PBS Tween 0.1% with mouse anti-PAR (1:1000). Proteins were revealed with fluorescent secondary anti-rabbit or anti-mouse antibodies (1:10 000; LI-COR) using the LI-COR fluorescent scanner or by chemiluminescence (Bio-Rad) using HRP-conjugated secondary anti-mouse antibody (2.5:10,000) and Amersham ECL prime detection reagent (GE healthcare lifescience).

## Immunostaining

For γH2AX immunofluorescence staining, $5 \times 10^4$ cells were seeded on glass slides (Thermo Fisher Scientific) and incubated overnight before adding treatments (Doxorubicin, carboplatin, olaparib). Forty-eight hr later, cells were fixed with formalin 10% (Sigma) for 10 min at room temperature, washed with PBS 1 X and then permeabilized with 0.25% Triton-X100 (Sigma) in PBS 1 X for 15 min. Slides were incubated 1 hr with DAKO blocking solution (Agilent) and incubated overnight at 4 C with anti-phospo-H2AX (1:2000). Cells were then washed three times with PBS 1X-Tween 0.05% followed by incubation with the secondary antibody (1:800; donkey anti-mouse IgG Alexa Fluor 488; Thermo Fisher Scientific). The glass slides were stained with DAPI, washed, and mounted with Fluoromount aqueous mounting medium (Sigma). Stained sections were scanned (Leica) by the molecular pathology core of the CRCHUM and photographs were analyzed using Visomorph viewer software for automatic counting of γH2AX foci.

## Data representation and statistical analysis

Mean +/- SEM are shown. Each experimental conditions for all independent experiments were tested in duplicates or triplicates excepted for XFe96 extracellular flux analysis (n=4–6 per condition tested). The numbers of repeated independent experiments are indicated in figure legends. Student T-test were performed when comparing 2 conditions. 1-way ANOVA was used when comparing 3 or more conditions. Statistical significance is shown in comparison to the control condition (untreated or CD73+), unless indicated otherwise by a bracket. Results are considered significant when $p<0.05$. All statistical analyses were performed using graph pad prism software (version 9.0.2).

## Acknowledgements

The authors thank Julien Lamontagne, Alexia Grangeon, Maxime Cahuzac, Dominique Gauchat, Daina Avizonis and Bozena Samborska for technical assistance.

# Additional information

### Competing interests

John Stagg: is permanent member of the Scientific Advisory Board and owns stocks of Surface Oncology, is member of the Scientific Advisory Board of Tarus Therapeutics, and is a member of the Scientific Advisory Board of Domain Therapeutics. The other authors declare that no competing interests exist.

## Funding

| Funder | Grant reference number | Author |
|---|---|---|
| Canadian Institutes of Health Research | | John Stagg |
| Fonds de recherche du Québec | | John Stagg |

The funders had no role in study design, data collection and interpretation, or the decision to submit the work for publication.

## Author contributions

David Allard, Conceptualization, Resources, Data curation, Formal analysis, Supervision, Validation, Investigation, Visualization, Methodology, Writing – original draft; Isabelle Cousineau, Resources, Formal analysis, Investigation, Methodology, Project administration; Eric H Ma, Hubert Fleury, Conceptualization, Resources; Bertrand Allard, Data curation, Formal analysis, Investigation, Methodology; Yacine Bareche, Data curation, Software, Visualization, Methodology; John Stagg, Conceptualization, Supervision, Funding acquisition, Writing – original draft, Project administration, Writing – review and editing

## Author ORCIDs

John Stagg http://orcid.org/0000-0001-7833-4228

## Ethics

All the experimental procedures were authorized, and all animals were handled according to an approved Institutional Animal Care and Use Committee (IACUC) protocol (#C20010JSs) of the Centre de Recherche du Centre Hospitalier de l'Université de Montréal.

## Decision letter and Author response

Decision letter https://doi.org/10.7554/eLife.84508.sa1
Author response https://doi.org/10.7554/eLife.84508.sa2

# Additional files

## Supplementary files

• Transparent reporting form

## Data availability

All data generated or analyzed during this study are included in the manuscript and supporting file.

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
