## [Editor Report]

This important study demonstrates a non-canonical, cancer-cell intrinsic role of the ectonucleotidase CD73 in the regulation of cancer cell metabolism. The evidence supporting the claims is convincing and of high quality.

---

## [Decision Letter]

**Decision letter after peer review:**

Thank you for submitting your article "The CD73 immune checkpoint promotes tumor cell metabolic fitness" for consideration by *eLife*. Your article has been reviewed by 2 peer reviewers, and the evaluation has been overseen by a Reviewing Editor and Anna Akhmanova as the Senior Editor. The following individual involved in the review of your submission has agreed to reveal their identity: Bassam Janji (Reviewer #2).

Essential revisions:

1) As outlined in the reviewers' reports, a number of validation-type experiments are needed. The authors should:

a. Measure the level of ADO released by CD73+, CD73- and CD73- gpASNase1 to support the functionality of CD73 in these cells and to fully endorse the results reported in Figure 2;

b. Directly test whether restoration of NAD+ levels restores aspartate synthesis;

c. Measure the rates of glutamine uptake in cells (e.g., consumption of glutamine from the media) in CD73+ vs CD73- cells over a time course;

d. Include the expression level of CD73 in wt MDA-MB-231 in Figure S1;

e. Measure OCR and ECAR levels of CD73+/- cells in nutrient-limited conditions to better correspond with functional assays;

f. Validate that exogenous NR used to restore mitochondria respiration is actually internalized;

g. Measure cell death versus proliferation occurring in the conditions used for the proliferation assays (e.g., Figure 1B, 1I, etc).

2) Additional experiments are needed to better establish the cause-effect type conclusions made in the manuscript:

a. To better evaluate the contribution of CD73 in the conversion of NMN to NR in MDA-MB-231, the effects of a CD73 blocking strategy (i.e. an anti-CD73 mAb) should be determined;

b. CD73+ or wt MDA-MB-231 should also be added in Figure 3D to see whether NR supplementation is sufficient to restore ECAR and OCR levels;

c. If the aim is to analyze the impact of NRK1 deletion on intracellular NAD synthesis, shGFP cells should be compared with shNRK1 cells, in both MDA-MB-231 CD73 pos and CD73 neg cells;

d. In addition to CD73, CD39 contributes to immune regulation through the hydrolysis of ATP to AMP which is a limiting step for extracellular adenosine production. Since CD73 catalyzes the final hydrolysis of AMP to adenosine following CD39 activity, the authors should evaluate whether the deletion of CD73 would also influence the expression of CD39 and A2AR.

3) The authors should edit the document in order to refine claims and/or increase the clarity of the results.

a. Change the text to "delay tumor growth" in Figure 1A

b. Add better descriptions of the nomenclature of isotopologues (M+1, M+2, etc) and the abbreviations for several of the seahorse assays

c. Tone down or adjust comments suggesting the role of CD73 in chemoresistance is elusive given the existence of extant literature, some from the authors' lab

d. Edit the results title "CD73 enhances aspartate biosynthesis to promote growth". As the loss of CD73 impairs aspartate biosynthesis, but it's unclear if CD73 enhances this process.

*Reviewer #1 (Recommendations for the authors):*

In this manuscript, "The CD73 immune checkpoint promotes tumor metabolic fitness" Allard et al. explore an immune-independent function of CD73 in cancer cells. Here, the authors demonstrate the cancer cell-intrinsic effects of CD73 to support cell growth and proliferation. Genetic ablation or pharmacological inhibition of CD73 impairs aspartate biosynthesis, oxidative phosphorylation, and impairs cell growth under nutrient-limiting conditions. CD73 converts extracellular nicotinamide mononucleotide to nicotinamide riboside, which generates intracellular NAD+. Lastly, the authors demonstrate that CD73 loss depletes intracellular NAD+, sensitizing these cells to DNA-damaging agents through decreased PARP activity.

Overall, the authors present a compelling case for the immune-independent roles of CD73 activity promoting cancer cell growth. The strengths of the paper are a thorough examination of CD73 inhibition in a series of cancer cell lines, including detailed metabolic studies and rescue experiments. However, several experimental points can be strengthened and clarified.

Comments

1) A key conclusion of this manuscript is that loss of CD73 leads to both decreased aspartate synthesis and decreased NAD+ levels. The authors state that restoration of NAD+ levels also rescues aspartate biosynthesis (Line 188), but no data is presented to support this claim- does restoration of NAD+ levels restore aspartate synthesis? If so, this would emphasize that CD73 generation of NAD+ is a key factor of metabolic regulation in this system.

2) Figure 1C, D. The authors state that there are no differences in cellular uptake of glucose or glutamine, but the evidence underlying this claim should be strengthened. In 1C, the authors should report the mean fluorescence intensity for 2-NBDG uptake in CD73- and + cells, as there does appear to be a shift between the two populations. In D, the authors compare relative levels of 13C-labelled glutamine. However, these values could be influenced by both uptake and intracellular metabolism (e.g., glutaminolysis) which cannot readily be distinguished by only measuring glutamine m+5. More evidence would be to measure the rates of glutamine uptake in cells (e.g., consumption of glutamine from the media) in CD73+ vs CD73- cells over a time course.

3) Several of the proliferation assays (e.g., Figure 1B, 1I, etc) culture cells in limiting nutrient levels for multiple days. Can the authors comment on the amount of cell death occurring across these conditions? Are some of these phenotypes due to differences in proliferation, or due to differences in cell death?

4) In Figure 1B, the authors demonstrate that impaired proliferation of CD73- cells occurs in nutrient-limiting conditions. Yet, the stable isotope tracing (Figure 1E, F) and oxygen consumption/ECAR assays (Figure 2) are all performed in nutrient-replete conditions. What are the OCR and ECAR levels of CD73+/- cells in nutrient-limited conditions? Is the decrease in aspartate biosynthesis exacerbated in nutrient-limiting conditions?

*Reviewer #2 (Recommendations for the authors):*

Although the authors showed that the metabolic function of CD73 is ADO/A2AR signaling independent, it is still important to measure the level of ADO released by CD73+, CD73- and CD73- gpASNase1 to support the functionality of CD73 in these cells and to fully endorse the results reported in Figure 2J.

To my best knowledge, MDA-MB-231 wt cells express high levels of CD73 (~ 90%). To evaluate whether the isolated CD73+ cells reflect the high level of CD73 expressed by their wt counterpart, it is important to include the expression level of CD73 in wt MDA-MB-231 in Figure S1.

In addition to CD73, CD39 contributes to immune regulation through the hydrolysis of ATP to AMP which is a limiting step for extracellular adenosine production. Since CD73 catalyzes the final hydrolysis of AMP to adenosine following CD39 activity, it would be interesting to evaluate whether the deletion of CD73 would also influence the expression of CD39 and A2AR.

In Figure 1A, there is a significant difference in the growth curves between CD73+ and CD73- tumors. However, we cannot appreciate on which day this significant difference appeared. Statistics should be done for each time point. The authors claim (line 92) that "CD73 deletion in MDA-MB-231 significantly suppressed tumor growth". This statement is not accurate, there is a significant delay in the tumor growth but not "suppression".

The authors should also describe what is the meaning of M, M+1, CR, SRC, etc in different figures.

Furthermore, it is important to determine whether the reduced ratios of ATP/ADP and ATP/AMP is in favor of an increase of ADP and AMP. Despite that the individual ADO receptor blockade and collective ADO receptor activation seem to show no involvement of ADO receptor activation in cell metabolism, the use of rhADA may not be a good strategy to avoid adenosine receptor activation since inosine is still able to bind adenosine receptors according to the literature (PMID: 26903141, 12947007).

To better evaluate the contribution of CD73 in the conversion of NMN to NR in MDA-MB-231, the use of anti-CD73 mAb should be performed (Figure 3B). In addition, if NR is a precursor of NAD, why the increase of extracellular NR is not associated with an increase in extra-cellular NAD?

The authors target intracellular NRK1 to inhibit intracellular NAD synthesis from NR, but what is the link between the levels of extracellular NMN and NR? This is not clear in the manuscript. Moreover, if the aim is to analyze the impact of NRK1 deletion on intracellular NAD synthesis, why not compare shGFP cells with shNRK1 cells, either for MDA-MB-231 CD73 pos or CD73 neg cells?

Figure 3A shows that NR supplementation restores normal intracellular NAD levels. This point needs to be discussed. CD73+ or wt MDA-MB-231 should also be added in Figure 3D to see whether NR supplementation is sufficient to restore ECAR and OCR to CD73+ or wt levels.

The authors claim that exogenous NR is used to restore mitochondria respiration, so they suppose that it is internalized but there is no data supporting that. Why NR should be used but not NMN?

Line 194: It is surprising that the authors consider that the mechanisms of CD73-mediated chemoresistance are elusive since the corresponding author of this manuscript published a PNAS paper describing that " CD73 overexpression in tumor cells conferred chemoresistance to doxorubicin by suppressing adaptive antitumor immune responses via activation of A2A adenosine receptors"

It would be interesting to comment on whether it is better to have CD73 and genomic stability or to delete CD73 and increase genomic instability, which may lead to more aggressive tumors.

---

## [Author Response]

Essential revisions:1) As outlined in the reviewers' reports, a number of validation-type experiments are needed. The authors should:a. Measure the level of ADO released by CD73+, CD73- and CD73- gpASNase1 to support the functionality of CD73 in these cells and to fully endorse the results reported in Figure 2;

As requested, we measured the levels of ADO released by CD73+, CD73- and CD73-gpASNase1 cells using a commercial assay based on malachite green measurement of inorganic phosphate resulting from the hydrolysis of exogenous AMP. We observed no residual CD73 enzymatic activity in either CD73- (ev) or CD73- gpASNase1 cells. We also measured CD73 protein expression by FACS and observed no impact of gpASNase1 on CD73 protein expression, consistent with the absence of enzymatic activity. These additional results have been added to our revised manuscript (new Figure 1—figure supplement 4).

b. Directly test whether restoration of NAD+ levels restores aspartate synthesis;

We thank the reviewers for this suggestion. We had shown that NR supplementation significantly restored intracellular NAD+ levels in CD73-deficient cells (see Figure 3A). To address the reviewers’ comment, we thus tested whether NR supplementation could effectively rescue aspartate synthesis in CD73-negative cells. Restoring NAD+ levels in CD73-negative cells with exogenous NR effectively restored aspartate synthesis to baseline levels. These new results have been added to our revised manuscript (new Figure 3F).

c. Measure the rates of glutamine uptake in cells (e.g., consumption of glutamine from the media) in CD73+ vs CD73- cells over a time course;

To address this question, we measured glutamine consumption in the conditioned media of CD73+ (CD73-proficient) and CD73- (CD73-deficient) cultures over a 3-days period. For this purpose, we used a commercial assay (Promega; #J8021) that measures glutamine levels. We observed that CD73 expression had no impact on glutamine consumption of MDA-231 cells. These new data are now presented in our revised manuscript (new Figure 1D).

d. Include the expression level of CD73 in wt MDA-MB-231 in Figure S1;

As requested, we have included CD73 surface expression analysis of the parental MDA-MB-231 cells. This new data is now presented in our revised manuscript (new Figure 1—figure supplement 1A).

e. Measure OCR and ECAR levels of CD73+/- cells in nutrient-limited conditions to better correspond with functional assays;

We thank reviewers for this suggestion. We have now performed OCR and ECAR measurements in cells cultured with 2 mM glutamine or 0.1 mM glutamine. We observed that reducing glutamine levels in the culture media diminished OCR of CD73+ cells, but had no impact on CD73- cells, which already displayed lower OCR levels. ECAR measurements were unaffected by glutamine concentrations. These new results suggest that CD73-proficient cells retain enhanced metabolic flexibility compared to CD73-deficient cells in glutamine-limiting conditions. We have added these results to our revised manuscript (new Figure 2—figure supplement 1). The text now reads as follow: “Strikingly, CD73neg MDA-MB-231 cells displayed significantly reduced OXPHOS (Figure 2B) and reduced glycolytic reserve compared to CD73pos cells in both glutamine replete and glutamine limiting conditions (Figure 2C, Figure 2—figure supplement 1).”

f. Validate that exogenous NR used to restore mitochondria respiration is actually internalized;

We agree with the reviewer that performing a tracing experiment with radiolabelled NR or NMN would strengthen our study. Unfortunately, despite our best efforts, we were unable to acquire isotopically labeled NR or NMN, and do not have the infrastructure to generate these in-house. We therefore cannot perform these experiments in due time. We therefore rely on previously published work (PMID: 27725675) to conclude that NR produced by CD73 is indeed internalized intracellularly.

g. Measure cell death versus proliferation occurring in the conditions used for the proliferation assays (e.g., Figure 1B, 1I, etc).

As suggested, we have measured cell death of our cells in nutrients-limiting conditions, using flow cytometry measurement of 7-AAD, a DNA dye that distinguishes viable and apoptotic/dead cells. We observed that CD73-deficient cells are more susceptible than CD73-proficient cells to low glutamine-induced cell death. We discussed these new results (new Figure 1—figure supplement 2D-E) in our revised manuscript. Text now reads as follow: “While NT5E (gene encoding for CD73) gene deletion had no impact in standard cell culture conditions (i.e. 25 mM glucose and 2 mM glutamine), it significantly suppressed cell proliferation and viability in nutrients-limiting conditions (Figure 1B, Figure 1—figure supplement 2), despite no difference in cellular uptake of glucose or glutamine, as shown using a fluorescent glucose analog (i.e. 2-NDBG; Figure 1C) and by measurements of glutamine depletion in media (Figure 1D).”

2) Additional experiments are needed to better establish the cause-effect type conclusions made in the manuscript:a. To better evaluate the contribution of CD73 in the conversion of NMN to NR in MDA-MB-231, the effects of a CD73 blocking strategy (i.e. an anti-CD73 mAb) should be determined;

We thank the reviewers for this suggestion. We have measured NMN-hydrolyzing activity of recombinant CD73 and CD73-expressing MDA-MB-231 cells in the presence and absence of the CD73 inhibitor APCP by LC-MS. We confirmed enzymatic activity of CD73 at performing hydrolysis of NMN into NR. These new results were added to our revised manuscript (new Figure 3—figure supplement 1).

b. CD73+ or wt MDA-MB-231 should also be added in Figure 3D to see whether NR supplementation is sufficient to restore ECAR and OCR levels;

We thank the reviewers for this suggestion. We have now added the OCR and ECAR profiles of CD73-proficient cells in parallel to CD73-deficient cells cultured with or without NR supplementation. A New Figure 3 was added to our revised manuscript (Figure 3D-E).

c. If the aim is to analyze the impact of NRK1 deletion on intracellular NAD synthesis, shGFP cells should be compared with shNRK1 cells, in both MDA-MB-231 CD73 pos and CD73 neg cells;

We have performed the statistical analysis and observed that NRK1-deficiency significantly impairs NAD+ biosynthesis in both CD73-proficient and deficient cells, albeit more importantly in CD73-proficient cells. The purpose of this experiment was to evaluate if NRK1 was required for CD73-mediated intracellular NAD synthesis. As such, a side-by-side comparison between CD73-proficient and deficient cells is more appropriate to conclude that CD73 contributes to NAD+ biosynthesis in a NRK1-dependent manner. Altogether, our data indicate that CD73 indeed contributes to NAD+ biosynthesis in NR-dependent and NRK1- dependent manner. Our New Figure 3 now includes statistical analyses comparing shGFP to shNRK1 for both CD73-proficient and CD73-deficient cells (New Figure 3C). We have added discussion to the results as follow: “NRK1-deficiency did reduce NAD+ levels in both CD73-proficient and CD73-deficient cells, suggesting that some intracellular NR may come from other sources (Figure 3C).”

d. In addition to CD73, CD39 contributes to immune regulation through the hydrolysis of ATP to AMP which is a limiting step for extracellular adenosine production. Since CD73 catalyzes the final hydrolysis of AMP to adenosine following CD39 activity, the authors should evaluate whether the deletion of CD73 would also influence the expression of CD39 and A2AR.

We have measured CD39 surface expression by FACS and found the cells do not express CD39. A2b is the most abundantly expressed ADO receptor in MDA-MB-231 cells (PMID: 27911956). We analyzed A2b and A2a gene expression by qPCR and found no significant differences in CD73-proficient and CD73- deficient cells. Results were added to our revised manuscript (new Figure 1—figure supplement 1).

3) The authors should edit the document in order to refine claims and/or increase the clarity of the results.a. Change the text to "delay tumor growth" in Figure 1A

We updated the figure title accordingly. Title of Figure 1 is now: “CD73-deficiency delays tumor growth independently from immune suppression through reduced aspartate biosynthesis.”

b. Add better descriptions of the nomenclature of isotopologues (M+1, M+2, etc) and the abbreviations for several of the seahorse assays

We apologize for the lack of clarification regarding certain abbreviations. We have added detailed description of abbreviations in respective figure legends of the revised manuscript.

c. Tone down or adjust comments suggesting the role of CD73 in chemoresistance is elusive given the existence of extant literature, some from the authors' lab

We have removed the comment suggesting the role of CD73 in chemoresistance is elusive, although we rather aimed at pointing that the complete mechanism of action is incompletely understood. The text now reads as follow: “Interestingly, early studies reported that tumor-derived CD73 can promote chemoresistance (1, 9, 10). To further our understanding of the mechanism of action by which CD73 regulates chemoresistance, we hypothesized that by increasing intracellular NAD levels, CD73 may regulate PARP activity and therefore promote genomic stability.”

d. Edit the results title "CD73 enhances aspartate biosynthesis to promote growth". As the loss of CD73 impairs aspartate biosynthesis, but it's unclear if CD73 enhances this process.

*We updated the title to “CD73-deficiency impairs aspartate biosynthesis and tumor growth” to* better reflect the conclusions.

Reviewer #1 (Recommendations for the authors):In this manuscript, "The CD73 immune checkpoint promotes tumor metabolic fitness" Allard et al. explore an immune-independent function of CD73 in cancer cells. Here, the authors demonstrate the cancer cell-intrinsic effects of CD73 to support cell growth and proliferation. Genetic ablation or pharmacological inhibition of CD73 impairs aspartate biosynthesis, oxidative phosphorylation, and impairs cell growth under nutrient-limiting conditions. CD73 converts extracellular nicotinamide mononucleotide to nicotinamide riboside, which generates intracellular NAD+. Lastly, the authors demonstrate that CD73 loss depletes intracellular NAD+, sensitizing these cells to DNA-damaging agents through decreased PARP activity.Overall, the authors present a compelling case for the immune-independent roles of CD73 activity promoting cancer cell growth. The strengths of the paper are a thorough examination of CD73 inhibition in a series of cancer cell lines, including detailed metabolic studies and rescue experiments. However, several experimental points can be strengthened and clarified.Comments1) A key conclusion of this manuscript is that loss of CD73 leads to both decreased aspartate synthesis and decreased NAD+ levels. The authors state that restoration of NAD+ levels also rescues aspartate biosynthesis (Line 188), but no data is presented to support this claim- does restoration of NAD+ levels restore aspartate synthesis? If so, this would emphasize that CD73 generation of NAD+ is a key factor of metabolic regulation in this system.

We thank Reviewer 1 for his/her constructive comments. In our original manuscript, we had shown that NR supplementation significantly restored intracellular NAD+ levels in CD73-deficient cells (see Figure 3A). To address the reviewers’ comment, we now tested whether NR supplementation could effectively rescue aspartate synthesis in CD73-negative cells. Restoring NAD+ levels in CD73-negative cells with exogenous NR effectively restored aspartate synthesis to baseline levels. These new results have been added to our revised manuscript (new Figure 3F).

2) Figure 1C, D. The authors state that there are no differences in cellular uptake of glucose or glutamine, but the evidence underlying this claim should be strengthened. In 1C, the authors should report the mean fluorescence intensity for 2-NBDG uptake in CD73- and + cells, as there does appear to be a shift between the two populations. In D, the authors compare relative levels of 13C-labelled glutamine. However, these values could be influenced by both uptake and intracellular metabolism (e.g., glutaminolysis) which cannot readily be distinguished by only measuring glutamine m+5. More evidence would be to measure the rates of glutamine uptake in cells (e.g., consumption of glutamine from the media) in CD73+ vs CD73- cells over a time course.

The fluorescence profiles of 2NBDG uptake overlap. This has been added to our revised manuscript (new Figure 1—figure supplement 2F).

Regarding glutamine uptake, we have now measured glutamine consumption in the conditioned media of CD73+ and CD73- cultures over a 4-days period using a commercial assay (Promega; #J8021). We observed that CD73 expression had no impact on glutamine consumption of MDA-231 cells. These new data are now presented in our revised manuscript (new Figure 1D).

3) Several of the proliferation assays (e.g., Figure 1B, 1I, etc) culture cells in limiting nutrient levels for multiple days. Can the authors comment on the amount of cell death occurring across these conditions? Are some of these phenotypes due to differences in proliferation, or due to differences in cell death?

As suggested, we have now measured cell death in nutrients-limiting conditions using flow cytometry measurement of 7-AAD, a DNA dye that distinguishes viable and apoptotic/dead cells. We observed that CD73-deficient cells are more susceptible to low glutamine-induced cell death. We discussed these new results (new Figure 1—figure supplement 2D-E) in our revised manuscript. Text now reads as follow: “While NT5E (gene encoding for CD73) gene deletion had no impact in standard cell culture conditions (i.e. 25 mM glucose and 2 mM glutamine), it significantly suppressed cell proliferation and viability in nutrients-limiting conditions (Figure 1B, Figure 1—figure supplement 2), despite no difference in cellular uptake of glucose or glutamine, as shown using a fluorescent glucose analog (i.e. 2-NDBG; Figure 1C) and by measurements of glutamine depletion in media (Figure 1D).”

4) In Figure 1B, the authors demonstrate that impaired proliferation of CD73- cells occurs in nutrient-limiting conditions. Yet, the stable isotope tracing (Figure 1E, F) and oxygen consumption/ECAR assays (Figure 2) are all performed in nutrient-replete conditions. What are the OCR and ECAR levels of CD73+/- cells in nutrient-limited conditions? Is the decrease in aspartate biosynthesis exacerbated in nutrient-limiting conditions?

We thank reviewers for this suggestion. We have now performed OCR and ECAR measurements in cells cultured with 2 mM glutamine or 0.1 mM glutamine. We observed that reducing glutamine levels in the culture media diminished OCR of CD73+ cells, but had no impact on CD73- cells, which already displayed lower OCR levels. ECAR measurements were unaffected by glutamine concentrations. These new results suggest that CD73-proficient cells retain enhanced metabolic flexibility compared to CD73-deficient cells in glutamine-limiting conditions. We have added these results to our revised manuscript (new Figure 2—figure supplement 1). The text now reads as follow: “Strikingly, CD73neg MDA-MB-231 cells displayed significantly reduced OXPHOS (Figure 2B) and reduced glycolytic reserve compared to CD73pos cells in both glutamine replete and glutamine limiting conditions (Figure 2C, Figure 2—figure supplement 1).”

Reviewer #2 (Recommendations for the authors):Although the authors showed that the metabolic function of CD73 is ADO/A2AR signaling independent, it is still important to measure the level of ADO released by CD73+, CD73- and CD73- gpASNase1 to support the functionality of CD73 in these cells and to fully endorse the results reported in Figure 2J.

As requested, we measured the levels of ADO released by CD73+, CD73- and CD73-gpASNase1 cells using a commercial assay based on malachite green measurement of inorganic phosphate resulting from the hydrolysis of exogenous AMP. We observed no residual CD73 enzymatic activity in either CD73- (ev) or CD73- gpASNase1 cells. We also measured CD73 protein expression by FACS and observed no impact of gpASNase1 on CD73 protein expression, consistent with the absence of enzymatic activity. These additional results have been added to our revised manuscript (new Figure 1—figure supplement 4).

To my best knowledge, MDA-MB-231 wt cells express high levels of CD73 (~ 90%). To evaluate whether the isolated CD73+ cells reflect the high level of CD73 expressed by their wt counterpart, it is important to include the expression level of CD73 in wt MDA-MB-231 in Figure S1.

As requested, we have included CD73 surface expression analysis of the parental MDA-MB-231 cells. This new data is now presented in our revised manuscript (new Figure 1—figure supplement 1A).

In addition to CD73, CD39 contributes to immune regulation through the hydrolysis of ATP to AMP which is a limiting step for extracellular adenosine production. Since CD73 catalyzes the final hydrolysis of AMP to adenosine following CD39 activity, it would be interesting to evaluate whether the deletion of CD73 would also influence the expression of CD39 and A2AR.

We have measured CD39 surface expression by FACS and found the cells do not express CD39. A2b is the most abundantly expressed ADO receptor in MDA-MB-231 cells (PMID: 27911956). We analyzed A2b and A2a gene expression by qPCR and found no significant differences in CD73-proficient and CD73- deficient cells. Results were added to our revised manuscript (new Figure 1—figure supplement 1).

In Figure 1A, there is a significant difference in the growth curves between CD73+ and CD73- tumors. However, we cannot appreciate on which day this significant difference appeared. Statistics should be done for each time point. The authors claim (line 92) that "CD73 deletion in MDA-MB-231 significantly suppressed tumor growth". This statement is not accurate, there is a significant delay in the tumor growth but not "suppression".

We have run statistical tests on all timepoints of each tumor growth curves and have added stats to the graphs (Figure 1A-B). We also have updated the description of the data in the revised manuscript and replaced “suppressed” by “delayed” as suggested.

The authors should also describe what is the meaning of M, M+1, CR, SRC, etc in different figures.

We apologize for the lack of clarification regarding certain abbreviations. We have added detailed description of abbreviations in respective figure legends of the revised manuscript.

Furthermore, it is important to determine whether the reduced ratios of ATP/ADP and ATP/AMP is in favor of an increase of ADP and AMP. Despite that the individual ADO receptor blockade and collective ADO receptor activation seem to show no involvement of ADO receptor activation in cell metabolism, the use of rhADA may not be a good strategy to avoid adenosine receptor activation since inosine is still able to bind adenosine receptors according to the literature (PMID: 26903141, 12947007).

We have added the absolute ATP, ADP and AMP levels as Supplementary data of our revised manuscript (new Figure 2—figure supplement 3A). We observed a decreased in total ATP levels in CD73-deficient cells, and no differences in ADP and AMP levels.

We agree with the reviewer’s comment that inosine can activate adenosine signaling. However, our seahorse experiments with either SCH58261, PSB1115 and NECA clearly demonstrate that the adenosine signaling axis is not involved in the regulation of OCR or ECAR (Figure 2J-K). Depleting ADO with exogenous rhADA not only prevents adenosine signaling, but also abrogates any potential effect of adenosine uptake via nucleotide transporters. We have added clarification in the result section of our revised manuscript. The text now reads as follow: “To rule-out a potential role for other ADO receptors or ADO intracellular uptake, we treated MDA-MB-231 cells with recombinant adenosine deaminase (rhADA) or the pan-ADO receptor agonist NECA.”

To better evaluate the contribution of CD73 in the conversion of NMN to NR in MDA-MB-231, the use of anti-CD73 mAb should be performed (Figure 3B). In addition, if NR is a precursor of NAD, why the increase of extracellular NR is not associated with an increase in extra-cellular NAD?

We thank the reviewers for this suggestion. We have measured NMN-hydrolyzing activity of recombinant CD73 and CD73-expressing MDA-MB-231 cells in the presence and absence of the CD73 inhibitor APCP by LC-MS. We confirmed enzymatic activity of CD73 at performing hydrolysis of NMN into NR. These new results were added to our revised manuscript (new Figure 3—figure supplement 1).

Additionally, to contribute to NAD biosynthesis, it was demonstrated that NR has to be converted back to NMN intracellularly via NRKs enzymatic activity (PMID: 27725675). NMN can then be converted to NAD via NMNAT enzymatic activity. There is no evidence as far as we know that either NRKs or NMNAT enzymes are found extracellularly or secreted. Extracellular NR is thus believed to serve as a NAD precursor upon cellular uptake (PMID: 27725675).

The authors target intracellular NRK1 to inhibit intracellular NAD synthesis from NR, but what is the link between the levels of extracellular NMN and NR? This is not clear in the manuscript. Moreover, if the aim is to analyze the impact of NRK1 deletion on intracellular NAD synthesis, why not compare shGFP cells with shNRK1 cells, either for MDA-MB-231 CD73 pos or CD73 neg cells?

We apologize for any lack of clarity. In the extracellular space, NMN has to be converted into NR to contribute to intracellular NAD+. Only NR, not NMN, can be transported intracellularly where it is phosphorylated by NRK1 to promote intracellular NAD+ synthesis (demonstrated in PMID: 27725675). We clarified with an added reference in the Results section of our revised manuscript. The text now reads as follow: “Upon intracellular transport, NR must be phosphorylated by intracellular NRK1 to promote intracellular NAD synthesis (PMID: 27725675).” We also added a graphical abstract in our supplementary data to highlight the proposed model (new Figure 5).

Additionally, newly added statistical analyses show that NRK1-deficiency significantly impairs NAD+ biosynthesis in both CD73-proficient and deficient cells, albeit more importantly in CD73-proficient cells. The purpose of this experiment was to evaluate if NRK1 was required for CD73-mediated intracellular NAD synthesis. Altogether, our data indicate that CD73 contributes to NAD+ biosynthesis in NR-dependent and NRK1-dependent manner. Our New Figure 3 now includes statistical analyses comparing shGFP to shNRK1 for both CD73-proficient and CD73-deficient cells (New Figure 3C). We have added discussion to the results as follow: “NRK1-deficiency did reduce NAD+ levels in both CD73-proficient and CD73-deficient cells, suggesting that some intracellular NR may come from other sources (Figure 3C).”

Figure 3A shows that NR supplementation restores normal intracellular NAD levels. This point needs to be discussed. CD73+ or wt MDA-MB-231 should also be added in Figure 3D to see whether NR supplementation is sufficient to restore ECAR and OCR to CD73+ or wt levels.The authors claim that exogenous NR is used to restore mitochondria respiration, so they suppose that it is internalized but there is no data supporting that. Why NR should be used but not NMN?

We have updated the result sections of the revised manuscript, to include the metabolic profiling of CD73+ cells to the graphs in panels D and E (new Figure 3D-E) as suggested. We used NR instead of NMN as only NR, not NMN, can be transported intracellularly where it is phosphorylated by NRK1 to promote intracellular NAD+ synthesis (demonstrated in PMID: 27725675). We added a graphical abstract in our supplementary data to highlight the proposed model and clarify the link between CD73, NMN, NR and NAD (new Figure 5).

Line 194: It is surprising that the authors consider that the mechanisms of CD73-mediated chemoresistance are elusive since the corresponding author of this manuscript published a PNAS paper describing that " CD73 overexpression in tumor cells conferred chemoresistance to doxorubicin by suppressing adaptive antitumor immune responses via activation of A2A adenosine receptors"

We have previously shown that CD73 confers chemoresistance in an immune-dependent manner (Loi S, PNAS). As suggested, we have toned down our comment suggesting the role of CD73 in chemoresistance is elusive, to mention instead that the complete mechanism of action is incompletely understood. The text now reads as follow: “Interestingly, early studies reported that tumor-derived CD73 can promote chemoresistance (1, 9, 10). To further our understanding of the mechanism of action by which CD73 regulates chemoresistance, we hypothesized that by increasing intracellular NAD levels, CD73 may regulate PARP activity and therefore promote genomic stability.”

It would be interesting to comment on whether it is better to have CD73 and genomic stability or to delete CD73 and increase genomic instability, which may lead to more aggressive tumors.

The impact of CD73 on genomic stability *per se* would be indeed an interesting question to follow-up. We thank the reviewer for this suggestion.